

# Assessment of gap-filling techniques applied to satellite phytoplankton composition products for the Atlantic Ocean

Ehsan Mehdipour[1,2], Hongyan Xi[1], Alexander Barth[3], Aida Alvera-Azcárate[3], Adalbert Wilhelm[2], Astrid Bracher[1,4]

[1]Alfred Wegener Institute (AWI), Helmholtz Centre for Polar and Marine Research, Bremerhaven, Germany
[2]School of Business, Social & Decision Sciences, Constructor University, Bremen, Germany
[3]GeoHydrodynamics and Environment Research (GHER), University of Liège, Liège, Belgium
[4]Institute of Environmental Physics, University of Bremen, Bremen, Germany

*Correspondence to*: Ehsan Mehdipour (ehsan.mehdipour@awi.de)

**Abstract.** Phytoplankton are vital to marine biogeochemical cycles and form the base of the marine food web. Comprehensive datasets offering a spatiotemporal perspective on phytoplankton composition are essential for assessing the impacts of climate change on marine ecosystems. Phytoplankton functional types (PFTs) classify phytoplankton based on their biogeochemical functions, enabling assessments of nutrient cycling, primary productivity, and ecosystem structure. However, satellite-derived ocean colour products like PFTs chlorophyll-a (Chla) concentrations are challenged by limited temporal and spatial coverage due to the exclusion of data collected under non-optimal observing conditions such as strong sun glint, clouds, thick aerosols, straylight, and large viewing angles or due to the specific sensor configuration and sensor malfunction. This highlights the importance of gap-filling techniques for producing consistent datasets, which are currently missing for operational data sets. This study evaluates two robust gap-filling methods for satellite observations: Data Interpolating Empirical Orthogonal Functions (DINEOF) and Data Interpolating Convolutional Auto Encoder (DINCAE). These methods were applied to Sentinel 3A/B OLCI-derived Chla concentration products in several regions of the Atlantic Ocean over three years of data, including total chlorophyll-a (TChla) and Chla concentration of five major PFTs, namely diatoms, dinoflagellates, haptophytes, green algae, and prokaryotic phytoplankton. The reconstructed datasets were assessed using test dataset evaluation and validated with in situ measurements collected during the transatlantic RV Polarstern expedition PS113 in 2018. The test dataset evaluation indicates that DINCAE outperforms DINEOF, particularly in capturing transient-scale features. DINCAE achieves an average root-mean-square-logarithmic-error (RMSLE) in cross-validation that is 66 % lower for TChla and 16 % lower for PFTs compared to DINEOF. However, external validation using in situ measurements indicates better performance for DINEOF than DINCAE, with improved regression metrics for PFTs, including a 12.5% better slope, 13.6% better intercept, and 68% higher coefficient of determination ($R^2$). The gap-filled datasets exhibit slightly reduced but still robust accuracy compared to the original satellite data while preserving statistical trends, improving spatial structure restoration, and increasing matchup data for validation. It is concluded that DINCAE and DINEOF each have unique strengths for gap-filling ocean colour products. DINCAE performs well in complex water bodies,





effectively reproducing patterns from the original satellite product. In contrast, DINEOF shows higher overall reliability, supported by independent validation, and is better suited for larger areas due to its lower computational demands.

**Keywords.** Phytoplankton functional types, gap-filling, DINCAE, DINEOF, validation, Sentinel 3A/B OLCI

**Graphical Abstract.**

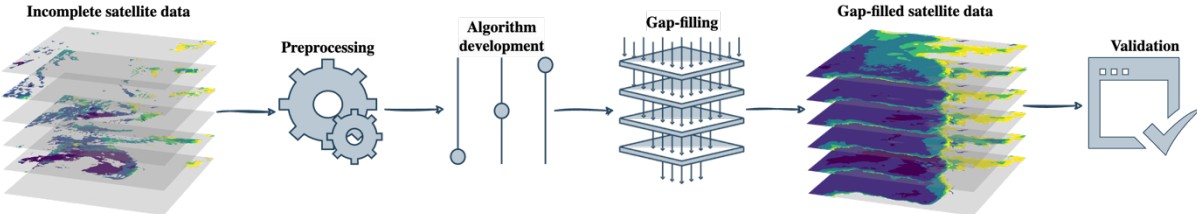

**Short Summary.** Phytoplankton are vital for marine ecosystems and nutrient cycling, detectable by optical satellites. Data gaps caused by clouds and other non-optimal conditions limit comprehensive analyses like trend monitoring. This study evaluated DINCAE and DINEOF gap-filling methods for reconstructing chlorophyll-a datasets, including total chlorophyll-a
and five major phytoplankton groups. Both methods showed robust reconstruction capabilities, aiding pattern detection and long-term ocean colour analysis.

## 1 Introduction

Phytoplankton are fundamental to marine biogeochemical cycles and ecosystems, contributing approximately 50 % of global primary production and providing over 90 % of the nutritional requirements for higher trophic levels within marine
ecosystems (Field et al., 1998). Understanding the spatiotemporal distribution of phytoplankton is crucial for assessing the impacts of climate change on ocean biogeochemistry, the marine food web, and the feedback mechanisms influencing oceanic and atmospheric processes (Fennel et al., 2019). Ocean colour remote sensing has significantly advanced our understanding of marine processes by providing continuous global data on surface chlorophyll-a (Chla) concentrations, a key indicator of phytoplankton biomass (Sathyendranath et al., 2019). In contrast to conventional chemical and biological
techniques, such as high-performance liquid chromatography (HPLC) and genetic analyses, optical methods facilitate broader and more consistent data acquisition. However, while Chla reflects overall phytoplankton biomass, it does not distinguish between different phytoplankton groups, each of which has unique morphological and physiological characteristics that contribute differently to biogeochemical cycles (Quéré et al., 2005).

Continuous monitoring of phytoplankton composition is critical for advancing our understanding of key biogeochemical
processes (Litchman et al., 2015), such as nutrient cycling (Arrigo, 2005) and carbon transfer (Basu and Mackey, 2018; Tréguer et al., 2018). This monitoring is also essential for managing fisheries (Stock et al., 2017), assessing water quality, and safeguarding public health (Garmendia et al., 2013), particularly concerning harmful algal blooms (Anderson, 2009).





Total chlorophyll-a (TChla) is a reliable indicator of total phytoplankton biomass in aquatic ecosystems (Huot et al., 2007), and it is widely used to monitor phytoplankton growth and blooms (Blondeau-Patissier et al., 2014). Despite its widespread

use, TChla provides a limited perspective since it does not capture the diversity and variability of the planktonic community structure. This limitation emphasizes the need for complementary metrics or approaches to better understand community dynamics and environmental interactions (Bracher et al., 2017). The significant role of various phytoplankton groups in marine food webs and biogeochemical cycles has spurred extensive research into their community structure and taxonomic composition (Bindoff et al., 2019). Understanding these aspects is vital for enhancing models that predict the impacts of

climate change on oceanic biogeochemical and ecological processes (e.g. Falkowski et al., 2003; Quéré et al., 2005).

Phytoplankton functional types (PFTs) are typically defined as groups of organisms linked by shared biogeochemical processes, such as silicification, calcification, and nitrogen fixation, though they may not be phylogenetically related (Falkowski et al., 2003; IOCCG, 2014; Litchman et al., 2006). Since many phytoplankton groups that are identifiable through remote sensing also function as PFTs (Bracher et al., 2017), these satellite-detected proxies are often referred to as

PFTs for simplicity (e.g. Losa et al., 2017). Among the PFTs, diatoms are recognized as major silicifiers and play a significant role in transporting carbon, nitrogen, and silica to deeper ocean layers (Armbrust, 2009; Nelson et al., 1995). Dinoflagellates are a phytoplankton group known for their ability to move actively within the water column, enabling them to optimize light and nutrient absorption. Some species of dinoflagellates are primary contributors to the most severe harmful algal blooms (HABs) in marine environments, causing significant harm to ecosystems and posing risks to human health

(IOCCG, 2014; Smayda, 1997). Haptophytes are a diverse group of phytoplankton that play a crucial role in marine ecosystems, particularly in the cycling of carbon and sulfur (Jordan and Chamberlain, 1997; Malin et al., 1992). A subgroup of them, coccolithophores, are known for their ability to form calcium carbonate shells, contributing significantly to the ocean's carbon sequestration processes (Rost and Riebesell, 2004). Green algae, also known as chlorophytes, contribute substantially to primary production in aquatic ecosystems. They play a vital role in carbon fixation and serve as a

foundational food source for various marine and freshwater organisms (Arora and Sahoo, 2015). Prokaryotic phytoplankton (hereafter referred to as prokaryotes), such as cyanobacteria or *Prochlorococcus* spp., are classified as picophytoplankton and are among the most abundant photosynthetic organisms in the ocean. Certain species, such as *Trichodesmium* spp., are particularly important due to their role in nitrogen fixation (Capone et al., 2005).

Satellite-derived Chla data are among the most effective datasets for investigating long-term variability in phytoplankton

communities (e.g. Sathyendranath et al., 2019). However, ocean colour data are often hindered by limited spatiotemporal coverage, primarily due to sensor constraints such as spectral, spatial, and temporal resolution, as well as interference from cloud cover, sun glint, dense aerosols, and other limitations (IOCCG, 2010; Steinmetz et al., 2011). Consequently, missing data presents substantial challenges in using satellite-derived biogeochemical parameters like Chla (Stock et al., 2020) for applications such as monitoring ecosystem dynamics, detecting seasonal and interannual variability, and modelling

biogeochemical processes. Several techniques are employed to address missing data for oceanic satellite products. Spatial interpolation methods such as Inverse Distance Weighting (IDW), Kriging (e.g. Kostopoulou, 2021), spline interpolation,



and nearest neighbour interpolation are used to estimate missing values based on spatial proximity (Li and Heap, 2008, 2014). Temporal interpolation or univariate interpolation methods like linear interpolation, polynomial interpolation, and spline interpolation aim to fill missing values within a time series of observations (Kandasamy et al., 2013; Lepot et al.,

2017). Spatiotemporal methods such as Optimal Interpolation (OI) (Hosoda and Sakaida, 2016; Reynolds and Smith, 1994) and data assimilation techniques like the Ensemble Kalman filter (Evensen, 2009; Nerger and Hiller, 2013) integrate spatial and temporal information.

Empirical and statistical methods such as empirical orthogonal function (EOF) (Alvera-Azcárate et al., 2005; Beckers and Rixen, 2003; Sirjacobs et al., 2011) analysis offer alternative strategies for gap-filling depending on the data characteristics

and intended application. Beckers and Rixen (2003) developed a method to estimate missing information using the EOF basis derived from the data. Building on this, Alvera-Azcárate et al. (2005) introduced the Data Interpolating Empirical Orthogonal Functions (DINEOF) method, a self-consistent technique based on EOF. This method is designed to fill in missing data within geophysical datasets using pre-existing spatiotemporal patterns, making it particularly effective for handling situations such as cloud-covered regions in satellite imagery or data interruptions caused by satellite malfunctions

(Beckers and Rixen, 2003). Comparative analysis with other spatiotemporal methods like OI reconstruction reveals outstanding similarities in results, with the DINEOF technique notably reducing processing time (Alvera-Azcárate et al., 2005). Unlike OI, however, this method operates independently of prior knowledge regarding the error covariance matrix, making it inherently self-consistent and parameters-free. DINEOF employs cross-validation (hold-out validation) to objectively determine the number of statistically significant EOFs (Beckers and Rixen, 2003). Alvera-Azcárate et al. (2007)

present a multivariate approach for reconstructing missing data through extended EOF. Extended EOFs represent an enhanced version of classical EOFs, involving the simultaneous use of multiple datasets for EOF analysis. The incorporation of multivariate EOFs proves advantageous in refining the analysis of a specific variable by incorporating related physical variables. This approach accounts for the correlation between variables through EOFs, enabling examination of their relationships and using this correlation to reconstruct the missing data. Furthermore, Alvera-Azcárate et al. (2009) applied

Laplacian filtering to the temporal covariance matrix before calculating EOFs, which helped reduce spurious variability and produced more realistic reconstructions. Stock et al. (2020) evaluated the performance of various gap-filling techniques, including Kriging, ridge regression, random forest, and DINEOF. Their findings demonstrated that DINEOF consistently ranked among the best-performing methods for gap-filling marine satellite products. However, linear approaches assume that variability can be represented through linear combinations of dominant modes, which limits their effectiveness in gap-filling

oceanic datasets; Due to the complex and non-linear interactions present in oceanic systems, these methods often struggle to reconstruct transient-scale structures and non-linear patterns accurately.

Machine learning gap-filling methods such as artificial neural networks (ANN) (Hong et al., 2023; Krasnopolsky et al., 2016), random forests (Park et al., 2019; Stock et al., 2020) , and self-organizing maps (SOM) (Abdel Latif et al., 2008; Chapman and Charantonis, 2017; Jouini et al., 2013) are designed to learn non-linear patterns within datasets to predict

missing values. These techniques present a promising approach for preserving transient-scale structures during data



reconstruction due to their ability to handle non-linear relationships and complex interactions. Among these, the Data Interpolating Convolutional Auto-Encoder (DINCAE) method represents a deep learning approach that was first introduced by Barth et al. in 2020 as DINCAE 1.0 and later enhanced in 2022 as DINCAE 2.0. This algorithm uses a neural network with a convolutional auto-encoder structure to reconstruct missing data from satellite observations using available cloud-free

pixels while also providing an error estimate for the reconstruction (Barth et al., 2020, 2022). Han et al. (2020) successfully applied DINCAE and DINEOF to reconstruct TChla in the South China Sea and the West Philippine Sea using SST and TChla data. Barth et al. (2021) used DINCAE to reconstruct TChla and total suspended particulate matter (SPM) in the Southern North Sea. Ji et al. (2021) employed DINCAE to reconstruct multi-source satellite data for TChla and SST in the East China Sea, investigating daily sea surface responses to typhoons. Jung et al. (2022) used DINCAE to reconstruct daily

SST from multi-satellite data sources and enhanced the results with data fusion of in situ measurement using a random forest approach. Luo et al. (2022) also applied the DINCAE method to reconstruct TChla data in the Bohai Sea and the Yellow Sea, comparing the results with those from DINEOF.

Understanding the distribution and dynamics of phytoplankton groups is essential for capturing the complete biogeochemical processes in the ocean. However, current Chla products do not adequately represent the dynamics of different phytoplankton

groups, and significant gaps exist in PFT datasets due to sensor limitations and cloud cover. Gap-free PFT datasets are currently unavailable due to a variety of reasons. First, significant gaps in satellite data exist due to factors such as cloud cover, sensor limitations, and limited spatiotemporal coverage (as PFT products rely on single-satellite observations). Second, closing these gaps necessitates advanced computational techniques, which are often resource-intensive and difficult to implement effectively, especially for large-scale datasets. Finally, the scarcity and uneven distribution of in situ

measurements, which are critical for ensuring the accuracy and reliability of gap-filled data, limit the validation of reconstructed datasets. These challenges make it difficult to generate consistent, high-quality gap-filled PFT datasets. This study aims to address these limitations by evaluating the performance of two well-established gap-filling methods in reconstructing TChla and the Chla concentrations of five major PFTs, namely diatoms, dinoflagellates, haptophytes, green algae, and prokaryotes, in the Atlantic Ocean. (Hereafter, for the sake of brevity, TChla and PFT refer to the TChla

concentration and Chla concentration of each PFT, respectively).

Each reconstruction and gap-filling method possesses distinct strengths and is selected based on the specific demands and constraints of the task at hand. DINEOF and DINCAE were chosen for this study because they are particularly suited to oceanographic datasets, where maintaining spatial and temporal continuity is crucial, and their advanced gap-filling capabilities ensure higher quality and more reliable data reconstruction compared to alternative approaches. These methods

surpass traditional interpolation techniques, such as kriging or simple regression, which often struggle to preserve the dynamic consistency of ocean processes. DINEOF employs empirical orthogonal functions to reconstruct missing values by extracting dominant modes of variability, making it particularly effective for large-scale oceanographic variables with spatial coherence. This method is actively used for gap-filling ocean colour products within the Copernicus Marine Service, including monthly global TChla and daily regional products (e.g., Mediterranean and Black Sea) at ~4 km resolution (Volpe



et al., 2018). Additionally, it is applied in NOAA CoastWatch multi-sensor global products, such as ~9 km TChla, SPM, and diffuse attenuation coefficient Kd(490) (Liu and Wang, 2018). Conversely, DINCAE employs a deep learning approach to capture complex non-linear relationships, offering greater flexibility and accuracy, particularly for highly variable data. By incorporating both anomaly estimation and error estimation in its cost function, DINCAE provides reliable performance in both reconstruction and error quantification (Barth et al., 2020, 2022). The gap-filled satellite data from both methods are

comprehensively evaluated using the in situ data collected from the expedition RV *Polarstern* PS113 transatlantic expedition in 2018.

In the following sections, we present the materials and methods used in this study. Section 2 covers the data sources, preprocessing steps, and methodologies for DINEOF and DINCAE, including model optimisation and validation metrics. Section 3 presents and analyses the evaluation results, concluding with a discussion of the strengths and limitations of the

reconstruction methods.

## 2 Materials and methods

### 2.1 Datasets

This study focuses on a corridor of the RV *Polarstern* PS113 transatlantic expedition (Alfred-Wegener-Institut Helmholtz-Zentrum für Polar- und Meeresforschung, 2017; Strass, 2018), which traversed from the Patagonian shelf to the English

Channel between 10 May and 9 June 2018 (Bracher et al., 2020a). This study used datasets from two distinct sources: the ship-borne dataset, which served as the basis for validating and comparing the gap-filling results, and three years of satellite-derived datasets, upon which the gap-filling methods were applied.

### 2.1.1 In situ dataset

The in situ measurements used in this study were based on phytoplankton pigment concentrations measurement by high-

pressure liquid chromatography (HPLC) and published in Bracher et al. (2020b). 230 surface water samples were gathered during the expedition for subsequent laboratory analysis of the phytoplankton pigment composition. The TChla was calculated as the sum of several chlorophyll-a pigments (monovinyl chlorophyll-a, divinyl chlorophyll-a, chlorophyll-a allomers, chlorophyll-a epimers, and chlorophyllide-a). The PFT concentrations were derived using the diagnostic pigment analysis (DPA) method based on Vidussi et al. (2001) to derive phytoplankton size classes and further refined by Hirata et al.

(2011) to derive PFTS with updated pigment-specific weighting coefficients following Xi et al. (2023a). This method enables the determination of five PFT concentrations: diatoms, dinoflagellates, haptophytes, green algae, and prokaryotes (Bracher et al., 2020a). The distribution of HPLC TChla and PFTs is depicted in Figure S1.



### 2.1.2 Satellite dataset

The satellite PFT products were acquired from the Copernicus Marine Services website (https://marine.copernicus.eu) (E.U.
Copernicus Marine Service Information, Marine Data Store), covering a temporal range of three years from 25 April 2016 to
25 April 2019, and spanning spatially from 64 °W to 3 °E and 50 °S to 52 °N. The data are derived using the algorithm
developed by Xi et al. (2021, 2020) within the Copernicus Marine Service framework. This algorithm extracts global PFT
concentrations from merged ocean colour (OC) products or Sentinel-3 (S3) A/B Ocean and Land Colour Instrument (OLCI)
data, using an expanded pigment database to determine PFT-specific coefficients. The algorithm employs EOF
decomposition to reduce the dimensionality of remote sensing reflectance spectral signals, followed by a multi-linear
regression method to establish the PFTs. The regression coefficients are calibrated using matchups with PFTs derived from
in situ HPLC pigment data. Subsequently, the model is applied to the global remote sensing reflectance dataset to generate a
global PFT dataset. Currently, this product is used for long-term monitoring of global phytoplankton groups (Xi et al.,
2023b, 2024). The product and dataset IDs from Copernicus Marine Service are
OCEANCOLOUR_GLO_BGC_L3_MY_009_103 and cmems_obs-oc_glo_bgc-plankton_my_l3-multi-4km_P1D,
respectively. Our study specifically focuses on five PFTs [product names in the dataset]: diatoms [DIATO], dinoflagellates
[DINO], green algae [GREEN], haptophytes [HAPTO], and prokaryotes [PROKAR], alongside the Total Chlorophyll-a
[CHL]. To maintain consistency, these product names for TChla and PFTs are used as representative abbreviation symbols
in figures.

The TChla dataset was produced from merged OC products by integrating data from multi-satellite missions, including
SeaWiFS, MERIS, MODIS-A, MODIS-T, VIIRS-SNPP & JPSS-1, and OLCI-S3A & S3B, resulting in a reduced rate of
data gaps (average of 52 %) (Figure 1 a, d and f). Conversely, the Copernicus Marine Service's PFT products are derived
from a more limited number of sources, OLCI-S3A & S3B, and exhibit a much higher rate of missing data (average of 82 %)
compared to the TChla product (Figure 1 b, e, and f). Notably, the missing data rate for PFTs decreased toward the end of
the study period as Sentinel-3B became operational on 25 April 2018, thereby narrowing the gap between satellite tracks
(Figure 1 c and f). Spatial analysis of missing data rates (Figure 1 d and e) reveals that the highest rates of missing data for
both TChla and PFTs occur predominantly in tropical and high-latitude regions, where they reach up to 80-90 % for TChla
and 90-100 % for PFTs. This high rate of missing data is largely attributable to persistent cloud cover in these regions.



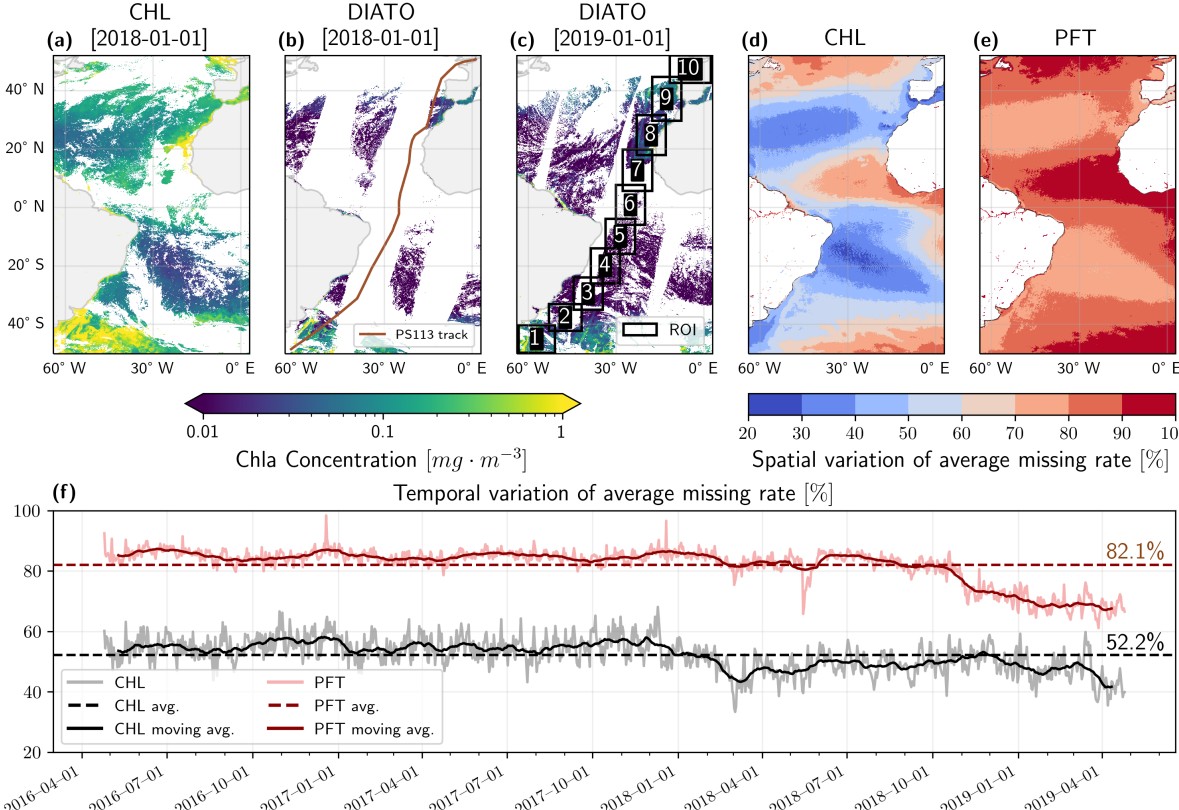

**Figure 1: Sample satellite datasets: (a) TChla on 1 January 2018 (b) diatoms on 1 January 2018 (including the PS113 expedition track), and (c) diatoms on 1 January 2019 (showing the ten extracted regions of interest, ROIs). Diatom is only used here as a representative example of all PFT products. (d–e) spatial variation of average missing rate for TChla and PFTs. (f) Temporal variation of average missing data rates for TChla and PFTs across the Atlantic Ocean. The light grey line represents the missing data rate for TChla, while the light red line indicates the missing rate for PFTs. Darker lines show the 30-day moving average, and the dotted dark lines denote the overall dataset's average missing rate.**

To assess the potential advantages of integrating an auxiliary dataset with no data gaps to enhance the gap-filling process in dataset reconstruction, the OSTIA foundation sea surface temperature (SST) dataset was used. This dataset, developed by GHRSST, the Met Office, and the Copernicus Marine Service (Donlon et al., 2012; Good et al., 2020), is catalogued in the Copernicus Marine Service under product ID SST_GLO_SST_L4_REP_OBSERVATIONS_010_011 and dataset ID METOFFICE-GLO-SST-L4-REP-OBS-SST. It is a Level 4 daily product provided at a spatial resolution of 0.05 degrees that was interpolated to the PFT product locations.

To assess the potential advantages of integrating an auxiliary dataset with no data gaps to enhance the gap-filling process in dataset reconstruction, the OSTIA foundation sea surface temperature (SST) dataset was used. This dataset, developed by GHRSST, the Met Office, and the Copernicus Marine Service (Donlon et al., 2012; Good et al., 2020), is catalogued in the Copernicus Marine Service under product ID SST_GLO_SST_L4_REP_OBSERVATIONS_010_011 and dataset ID



METOFFICE-GLO-SST-L4-REP-OBS-SST. It is a Level 4 daily product provided at a spatial resolution of 0.05 degrees that was interpolated to the PFT product locations.

## 2.2 Satellite data preprocessing

Satellite datasets require careful preprocessing before they can be used in gap-filling models. This process involves restructuring and reformatting the data to meet specific model requirements and assumptions, such as the normality of input datasets. These steps are essential for enhancing model performance and ensuring the reliability of the gap-filling results. The preprocessing of satellite data prior to input into the gap-filling models involved the following steps: (1) normalization of the log-normally distributed Chla datasets, (2) extraction of regions of interest within the corridor surrounding the PS113 expedition, and (3) partitioning of the dataset into training, validation and test datasets using an artificial cloud mask. Figure 2 illustrates the data processing workflow, providing an overview of the sequential steps and methodologies employed in the analysis.




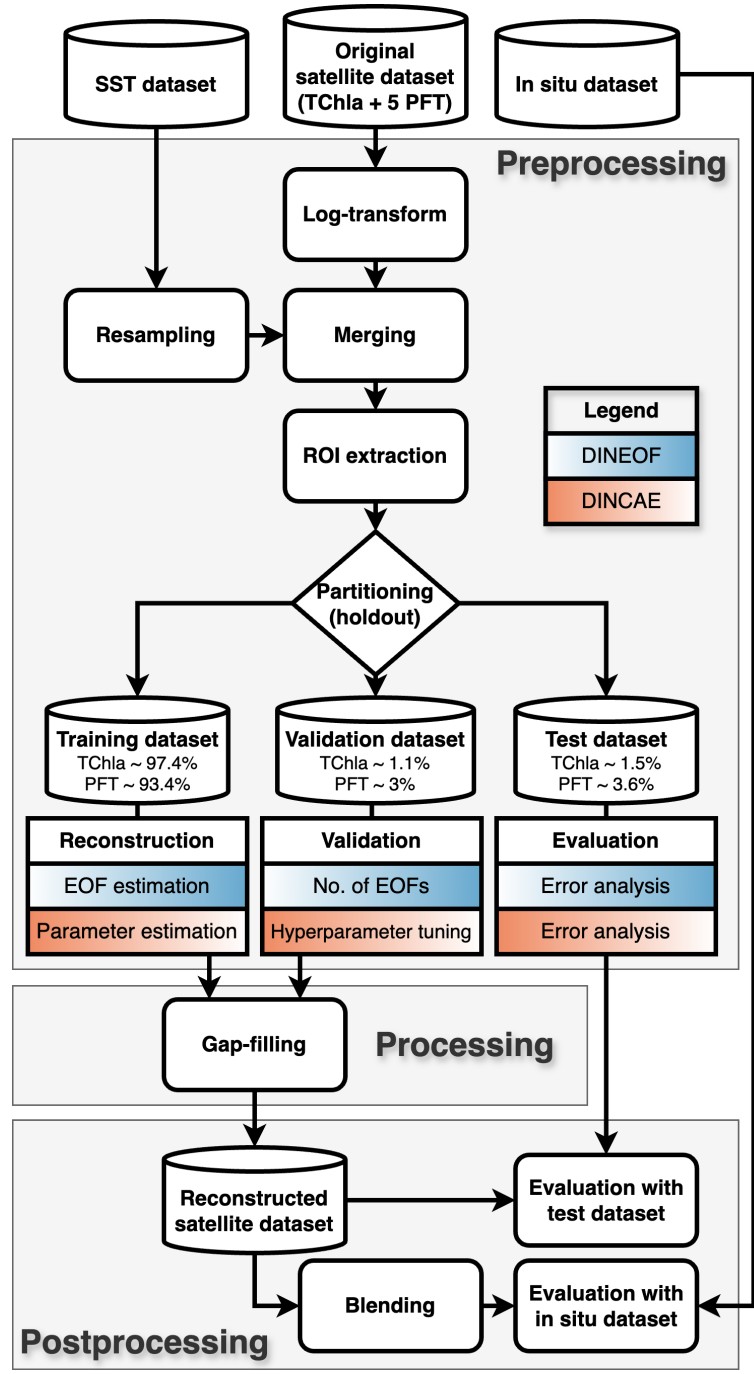

**Figure 2: Diagram of the data processing workflow, divided into three sections: Preprocessing, which includes extracting regions of interest (ROIs) and partitioning the dataset into training, validation, and test subsets; Processing, which encompasses model configuration development and gap-filling across all areas; and Postprocessing, which includes validation of the satellite-derived products using the holdout test dataset and in situ measurements.**

245



### 2.2.1 Normalization

Phytoplankton Chla typically follows a log-normal distribution, influenced by the multiplicative effects of environmental factors like light, nutrients, and temperature on their growth (Campbell, 1995). Consequently, employing log-transformed Chla concentration in modelling proves more effective, as it normalizes the skewed distribution, diminishes the impact of outliers, and more accurately reflects the underlying ecological processes. Log-transformation also ensures that gap-filling results in positive Chla concentrations.

### 2.2.2 Region of interest (ROI) extraction

Due to the substantial computational demand and inherent scalability limitations of both algorithms, direct reconstruction of the dataset for the whole Atlantic Ocean was deemed unfeasible. Efforts to process the entire region restricted the DINEOF algorithm to approximately four months of data, resulting in fewer EOF extractions and a lower-quality reconstruction. Furthermore, the DINCAE algorithm experienced GPU memory crashes when processing the entire dataset. To address these challenges, similar to Jung et al. (2022), the satellite dataset was divided into smaller, more computationally manageable areas (tiles) focused along a corridor of the expedition track, aiming to generate overlapping regions (1° buffer) and produce a continuous dataset that facilitates meaningful comparisons between the in situ measurements and satellite products, both before and after gap-filling. The challenge was to determine an optimal number of areas that would reduce the computational load while ensuring that each area was sufficiently large to accurately capture the dynamics of the region. The areas were designed to be comparable in size or larger than those used in previous gap-filling studies (Barth et al., 2020, 2022; Han et al., 2020; Jung et al., 2022). Using a k-means clustering algorithm, we classified the location of the research vessel during the PS113 expedition into ten areas, with each area covering at least ten degrees of longitude (Figure 1 c). This strategy ensured some data availability even when gaps between satellite tracks overlapped the area, thereby reducing the number of days without any data in PFT products. Figure S2 illustrates the boundaries of the regions of interest (ROI) overlaid on continental shelves (Flanders Marine Institute, 2023, 2024) and in situ measurement clustered by Bracher et al. (2020a) into the Longhurst biogeochemical provinces (Longhurst, 2010). Table 1 summarizes the physical and biogeochemical characteristics of the areas along with the major overlapping Longhurst provinces (Longhurst, 2010) and hierarchical clusters defined by Bracher et al (2020a).

The original satellite datasets initially spanned 1095 days across all regions. To improve the robustness of the reconstruction, days with less than 2 % data availability were excluded. Area No.7 experienced the highest proportion of missing data, largely due to its proximity to the West African upwelling region, a zone frequently obscured by cloud cover. As a result, only 885 days of data were retained for area No.7, while a minimum of 1048 days were preserved for the other regions. Following both gap-filling methods, finally a weighted blending method, commonly referred to as alpha blending or feathering, was employed to merge the areas (Lu et al., 2014; Uyttendaele et al., 2001). This technique ensures a smooth



transition in the overlapping areas (Figure 1 c) with the weight coefficient determined by the distance from the stitching

borders (Lu et al., 2014).

**Table 1: Summary of areas' physical and biogeochemical characteristics and major overlapping Longhurst provinces. SWAS for Southwest Atlantic Shelves, BRAZ for Brazilian Current Coast, SATL for South Atlantic Tropical Gyre, WTRA for Western Tropical Atlantic, NATR for North Atlantic Tropical Gyre, CNRY for Canary Current Coast, NASE North Atlantic Subtropical Gyre East, NASE-N for Northern NASE, NADR for North Atlantic Drift and NECS for Northeast Atlantic Shelves.**

| Area No. | Major overlapping Longhurst provinces clustered by Bracher et al. (2020a) | Major overlapping hierarchical clusters defined by Bracher et al. (2020a) | Biogeochemical characteristic | Physical characteristic |
|---|---|---|---|---|
| 1 | SWAS and BRAZ | I and V | Mesotrophic | Continental shelf |
| 2 | SATL | II | Oligotrophic | Continental shelf |
| 3 | SATL | II | Oligotrophic | Continental shelf |
| 4 | SATL | II | Oligotrophic | Continental shelf |
| 5 | SATL and WTRA | II | Oligotrophic | Open ocean |
| 6 | WTRA | II | Oligotrophic and partially mesotrophic | Equatorial open ocean |
| 7 | CNRY and WTRA | III and II | Eutrophic and mesotrophic | Continental shelf and upwelling (von Appen et al., 2020) |
| 8 | NATR, NASE, and CNRY | II and III | Eutrophic and mesotrophic | Coastal and continental shelf |
| 9 | NASE and NASE-N | II and IV | Eutrophic and mesotrophic | Coastal and continental shelf |
| 10 | NADR and NECS | IV and VI | Eutrophic | Coastal and continental shelf |

### 2.2.3 Data partitioning

The data from each area were divided into three distinct datasets: training, validation (development), and test, as illustrated in Figure 2. The training dataset was used to analyse the satellite data and estimate the internal parameters of each algorithm; specifically, estimating EOF patterns in the DINEOF algorithm and determining the weights in the DINCAE algorithm. The validation dataset served to fine-tune the algorithms for optimal performance and choosing the model architecture, such as identifying the optimal number of EOFs in DINEOF and adjusting the hyperparameters in the DINCAE gap-filling method.

The structure with the lowest error on the validation dataset was selected. Following this, the algorithms were retrained using a combination of the training and validation datasets with optimal configuration to acquire the gap-filled products. The test dataset provided independent data for evaluating both algorithms and comparing the performances. In the validation and test phase, the significance of TChla and all PFTs were considered equally when calculating the average errors. Therefore, the amount of masked data was similar across groups in each phase. However, differences in data availability resulted in varying



percentages of masked data (Figure 2). 1.1 % and 1.5 % of the TChla data and 3 % and 3.6 % of the PFTs data were allocated to the validation and test datasets, respectively. Typically, 1 % of the initial dataset is reserved for cross-validation, as noted by Alvera-Azcárate et al. (2007, 2009). Recent studies have employed higher percentages, such as 3 % in Wang et al. (2019), 2–3 % in Alvera-Azcárate et al. (2021), 5 % in Liu and Wang (2022), and 3 % in Alvera-Azcárate et al. (2024), aligning with the percentages used in our study.

The validation and test datasets were generated using a data partitioning technique similar to that described by Alvera-Azcárate et al. (2009), Barth et al. (2020) and Beckers et al. (2006). In this approach, cloud masks were extracted from cloudy days and subsequently applied to other days, effectively obscuring portions of the dataset for evaluation purposes. However, using the holdout method suggested by Barth et al. (2020) (where cloud masks were derived from the first 50 days and used to obscure random days or the last 50 days) our initial trials encountered issues, resulting in timestamps either

devoid of any remaining data or overlapping with pre-existing gaps, thereby preventing the formation of a valid evaluation dataset. To solve this issue, we adopted a method similar to Alvera-Azcárate et al. (2009) that better suits the characteristics of our dataset. The process for generating the validation and test datasets is depicted in Figure 3. The data for each area was grouped into monthly batches, where each batch corresponds to a specific month of the year (e.g., January or February) regardless of the year. This approach ensured an even distribution of validation and test datasets across all months, reducing

potential seasonal biases. Subsequently, daily maps with less than 2 % data coverage—calculated based on the combined matrix of TChla and PFTs—were excluded from the analysis to maintain data quality and consistency. To further refine the process, we identified the 10th percentile data completeness level in all timestamps and selected the first image beyond this threshold to serve as the cloud mask. This approach minimised the risk of mistaking satellite track gaps for cloud-induced gaps, ensuring that a small portion of the masked image was always preserved, and some patterns were retained to enable

more accurate reconstruction and evaluation. The cloud mask was applied exclusively to pixels missing in both TChla and PFTs, targeting the timestamp with the highest data availability within each month. By simulating realistic data gaps, this method provided a robust framework for evaluating the performance of gap-filling algorithms.

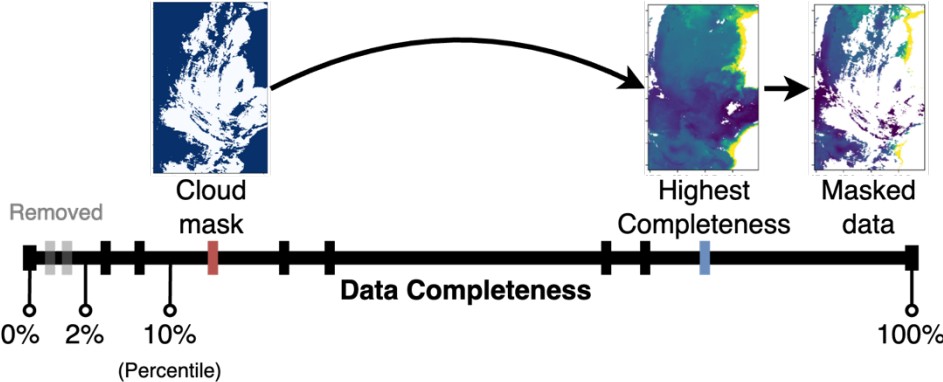

**Figure 3: Schematic representation of cloud mask creation and implementation. Grey bars represent timestamps with less than 2**
**% data availability, which were excluded from the reconstruction. The red bar denotes a timestamp with an adequate cloud**



**presence (not fully obscuring the region), used as the cloud mask. The blue bar represents the timestamp with the highest data availability, which was masked to generate validation or test datasets.**

### 2.3 Gap-filling methods

**2.3.1 DINEOF gap-filling**

We provide a brief overview of the DINEOF algorithms along with the key modifications applied in this study. More details can be found in Alvera-Azcárate et al. (2009, 2007, 2005) and Beckers and Rixen (2003). The DINEOF approach handles missing data through an iterative process. Initially, the method centres the data by subtracting the mean and replaces the missing values with zero. The EOF is then computed using this updated matrix, and the primary EOF is used to predict the

values at the locations where data was initially missing. This iteration continues until the anomalies at the missing values converge to a specified level from one iteration to the next. Once convergence is achieved, the number of calculated EOFs gradually increases, up to a maximum of $k_{max}$ EOFs, or stops if the cross-validation error rises continuously.

In our study, the input matrix was constructed by combining multiple datasets of TChla and five PFTs, resulting in $X_e$ matrix shown in (1. $X$ is a column vector containing all the spatial points of TChla and PFTs datasets across $T$ temporal steps. The

inclusion of the SST dataset was also tested to assess its impact on enhancing pattern recognition and improving reconstruction accuracy.

$$X_e = \begin{bmatrix} t_1 & t_2 & ... & t_T \\ X_{CHL} & X_{CHL} & ... & X_{CHL} \\ X_{DIATO} & X_{DIATO} & ... & X_{DIATO} \\ X_{DINO} & X_{DINO} & ... & X_{DINO} \\ X_{HAPTO} & X_{HAPTO} & ... & X_{HAPTO} \\ X_{GREEN} & X_{GREEN} & ... & X_{GREEN} \\ X_{PROKAR} & X_{PROKAR} & ... & X_{PROKAR} \end{bmatrix} \tag{1}$$

**2.3.2 DINCAE gap-filling**

DINCAE's architecture, similar to convolutional autoencoders and U-Net networks (Ronneberger et al., 2015), employs sequence of encoder-decoder layers to extract significant features from irregular and sparse data through dimensionality

reduction, similar to EOF methods (Jung et al., 2022). In this structure, input data is compressed through a bottleneck by the encoder using convolutional and average-pooling layers to reduce resolution, then decompressed by the decoder with convolutional and interpolation layers. Similar to a U-Net, DINCAE optionally incorporates skip connections, which allow some information to bypass the bottleneck and retain more of the original gradient features during reconstruction (Barth et al., 2020, 2022; Jung et al., 2022; Ronneberger et al., 2015). DINCAE employs a non-linear approach to OI using

convolutional operations, modelling the neural network output as a Gaussian probability distribution with a predicted mean




$\hat{y}_{ij}$ and expected error variance $\hat{\sigma}_{ij}^2$ for each grid point $i, j$. The weights and biases of the neural network are optimized to maximize the likelihood of the observed values $y_{ij}$. The corresponding cost function is formulated as described in Eq. (2.

$$J(\hat{y}_{ij}, \hat{\sigma}_{ij}^2) = \frac{1}{2N} \sum_{ij} \left[ \left( \frac{y_{ij} - \hat{y}_{ij}}{\hat{\sigma}_{ij}} \right)^2 + \log(\hat{\sigma}_{ij}^2) \right] \qquad (2)$$

Where $N$ is the number of non-masked data points in $y_{ij}$. The first term of the cost function is related to the mean square error, adjusted by the estimated error standard deviation. The second term penalizes overestimation of the error standard deviation. In DINCAE 2.0, a convolutional auto-encoder with refinement, the intermediate results $\hat{y}_{ij}$ and $\hat{\sigma}_{ij}^2$ are combined with the inputs and processed through another auto-encoder with a similar architecture and independent weights from the initial layer (Barth et al., 2022). The ultimate cost function incorporating refinement denoted as Jr, is expressed as:

$$J_r = \alpha J(\hat{y}_{ij}, \hat{\sigma}_{ij}^2) + \alpha' J(\hat{y}'_{ij}, \hat{\sigma}'^2_{ij}) \qquad (3)$$

Here, $\hat{y}'_{ij}$ and $\hat{\sigma}'^2_{ij}$ represent the reconstruction and the expected error variance generated by the second auto-encoder. The weights α and α' control the relative significance attributed to the intermediate and final outputs within the cost function, and they would be fine-tuned during hyperparameter optimization. Incorporating a refinement step effectively doubles the neural network's depth and nearly doubles the number of parameters, substantially increasing its complexity. This added depth enhances the network's ability to capture more intricate data patterns and relations but also results in higher computational costs. For more detailed information on the method, refer to Barth et al. (2022, 2020).

The main input variables for the DINCAE model include anomalies in input data and the inverse of error variance spanning consecutive days centred around the reconstruction day. Additionally, spatiotemporal coordinates were incorporated as auxiliary input variables. In our study, the number of input variables expanded due to the adoption of a multivariate approach and the inclusion of all PFT datasets. As outlined in Table 2, for a 3-day timeframe, there are 36 primary variables, with an additional 4 auxiliary variables, resulting in a total of 40 layers. The model's output consists of the reconstructed PFTs and the corresponding expected error variance of the reconstruction for each variable.

**Table 2: Example summary of input and output variables of DINCAE in reconstructing PFTs for the 3-day time window. Chla anomalies are the Chla concentrations of TChla and PFTs.**

| Variable type | Set of variables | Variable name |
|---|---|---|
| Input<br>Main variables | 6 ×<br>(TChla and 5 PFTs) | 1. Chla anomalies scaled by the inverse of the error variance (the scaled anomaly is zero when data are absent)<br>2. Inverse of the error variance (zero when data are absent)<br>3. Scaled Chla anomalies of the previous days<br>4. Inverse of error variance of the previous days<br>5. Scaled Chla anomalies of the next days<br>6. Inverse of error variance of the next days |
| Input | 1 × | 1. Longitude (scaled linearly between -1 and 1)<br>2. Latitude (scaled linearly between -1 and 1) |



| Auxiliary variables | | 3. Cosine of the day of the year with a period of 365.25 days<br>4. Sine of the day of the year with a period of 365.25 days |
| --- | --- | --- |
| Output | $6 \times$<br>(TChla and 5 PFTs) | 1. Reconstructed Chla product for TChla and each PFT scaled by the inverse of the expected error variance<br>2. Logarithm of the inverse of the expected error variance for each product |

### 2.3.3 Model development and hyperparameter optimization

Hyperparameter optimization is crucial for enhancing model performance. In contrast to model parameters, hyperparameters are defined prior to training and play a significant role in shaping the model's accuracy and behaviour. Their optimization is
typically conducted during the validation (development) step to ensure optimal model performance. Traditional methods like grid search explore a predefined set of hyperparameter values iteratively, but this approach can be computationally intensive and inefficient, especially with large datasets and complex models. Random search offers a more efficient alternative by randomly selecting hyperparameter combinations from a given distribution. Although seemingly counterintuitive, research has demonstrated that random search often outperforms grid search, particularly when only a few hyperparameters have a
major impact on model performance (Bergstra et al., 2011; Bergstra and Bengio, 2012; Goodfellow et al., 2016; Hutter et al., 2019). Although DINEOF is inherently a self-consistent algorithm, its temporal Laplacian filtering process requires the optimization of two key parameters to achieve optimal results. The parameter $\alpha$ governs the filter's intensity, while $p$ denotes the number of iterations during which the filter is applied to the temporal covariance matrix. The effective extent of the Laplacian filter is calculated as $L = 2\pi\sqrt{\alpha p}$. Further information related to the filtering method can be obtained from
Alvera-Azcárate et al. (2009). Additionally, random search includes a test condition to evaluate the algorithm's performance with and without SST in the dataset. In contrast, DINCAE involves numerous hyperparameters related to the input dataset, deep learning architecture, generalization, and cost function optimization. Table S1 and Table S2 outline the hyperparameters for DINEOF and DINCAE, respectively, including descriptions, selection ranges, distributions, and selected values.

Training individual models for each area demands significant computational resources. To optimize efficiency, hyperparameters were fine-tuned on a representative area and then generalized to other areas. Area No. 9 was selected for this purpose due to its dynamic features, including freshwater influx from coastal rivers and moderate phytoplankton levels, which enable the algorithm to better capture fluctuations compared to oligotrophic regions. Additionally, area No. 9 has a moderate rate of missing data, unlike areas with severe gaps, such as area No. 7, enhancing the algorithm's robustness. As
the largest area, it also ensures that computational demands for subsequent analyses remain manageable. DINEOF computations were executed on a system equipped with AMD Rome Epyc 7702 processors. Each DINEOF training run (a full cycle of model training) during the development phase averaged 17 hours, with durations ranging from 12 to 24 hours. DINCAE computations were carried out on an Nvidia A100 GPU supported by AMD Rome Epyc 7702 CPU. On average, each DINCAE training run during the development phase took approximately 15 hours, with durations ranging from 6 to 28



hours. The computation time for DINCAE is particularly sensitive to the number of time windows and epochs required for

model training.

## 2.4 Validation and evaluation metrics

The model's performance was assessed through holdout validation (termed cross-validation in literature) on the validation

dataset during the development and final evaluation of the test dataset, as described in Sect. 2.2 for data partitioning.

Additionally, two complementary approaches were employed to evaluate the performance of the gap-filling techniques.

First, the gradient field and degree of smoothing were assessed to qualitatively and quantitatively examine the impact of

reconstruction on the smoothing of the original satellite dataset. Second, mathematical and statistical metrics derived from

the validation regression analysis of matchups between in situ measurements and satellite-derived products were analysed,

using in situ measurements described in Sect. 2.1.1.

**2.4.1 Performance evaluation**

The optimal model during development is identified by the lowest root-mean-square-logarithmic-error (RMSLE) across

TChla and all PFTs ((4), with no bias towards any specific PFT, ensuring equal contribution in the total RMSLE calculation

using the same number of holdout validation points. Model performance is further evaluated by RMSLE, comparing the

reconstructed data to the test dataset. While RMSLE provides a useful comparison of different models on a logarithmic

scale, it is not easily interpretable. Therefore, relative errors metrics, particularly the mean-absolute-percentage-error

(MAPE) defined in (5 are used for a clearer assessment of final model performance. MAPE evaluates the reconstructed Chla

on a linear scale, offering a more intuitive percentage-based interpretation, particularly useful for skewed data distributions.

Unlike squared error metrics, MAPE uses absolute percentage values, which reduces the impact of outliers. However, when

original values are near zero, percentage errors can become disproportionately large and distorting the mean. To avoid this, a

small percentage (approximately 0.01 %) of extreme values (i.e., PE > 10,000 %) were excluded from MAPE calculations to

obtain a more robust assessment.

$$RMSLE = \sqrt{\frac{1}{M}\sum_{i=1}^{M}\left(log_{10}(C_{rec,i}) - log_{10}(C_{CV,i})\right)^2} \qquad (4)$$

$$MAPE = \frac{1}{M}\sum_{i=1}^{M}\left|\frac{C_{rec,i} - C_{CV,i}}{C_{CV,i}}\right| \times 100\ \% \qquad (5)$$



Where M and $C_{CV,i}$ refers to the number and values of the holdout validation points (i.e. validation or test dataset) respectively and $C_{rec,i}$ is the reconstructed value.

### 2.4.2 Spatial smoothing

The Sobel edge operator is extensively used for edge detection in image fields (Sobel and Feldman, 1968). This operator relies on a 3x3 convolution mask, consisting of a horizontal and vertical kernel, uniformly applied across the dataset to effectively extract gradient changes (Vincent and Folorunso, 2009). The magnitude of the gradient field is widely used in oceanography to define water mass boundaries and analyse dynamic changes in oceanic structures, help in understanding the spatial variability and the intensity of currents, fronts, and eddies (Belkin and O'Reilly, 2009; Wang et al., 2021). Although

this method is typically used to extract gradients on SST, this study uses the Sobel operator to identify gradients in the Chla dataset, aiming to qualitatively assess the performance of a gap-filling algorithm in reconstructing these gradients. The magnitude of the gradient field (G) is expressed as follows:

$$G_x = \begin{bmatrix} -1 & 0 & +1 \\ -2 & 0 & +2 \\ -1 & 0 & +1 \end{bmatrix} * S \cdot (2d_x)^{-1}$$

$$G_y = \begin{bmatrix} -1 & -2 & -1 \\ 0 & 0 & 0 \\ +1 & +2 & +1 \end{bmatrix} * S \cdot (2d_y)^{-1} \tag{6}$$

$$G = \sqrt{G_x^2 + G_y^2}$$

In this formula, the symbol * signifies the convolution operator, $S$ represents the source image undergoing processing, and $d_x$ and $d_y$ denote the resolution in the horizontal and vertical dimensions, respectively. These distances are used to normalize

the gradient values, accounting for the physical spacing between pixels in each direction. Normalization may be omitted if there is no need to adjust gradient magnitudes for pixel spacing, particularly when the focus is on visually assessing gradient changes. For this analysis, a pixel size of 4 km was applied, consistent with the nominal resolution of the TChla and PFT products. In our study, the gradient is expressed in units of $mg \cdot m^{-3} \cdot m^{-1}$, which simplifies to $mg \cdot m^{-4}$.

The degree of smoothing reflects how closely reconstructed data aligns with the original input, with less smoothing

indicating better preservation of the original values. This is quantified using the RMSLE, similar to Eq. (*4* but applied to the differences between all data from the training and validation datasets and the reconstructed dataset. Here, a lower RMSLE indicates better preservation of the original data's details. However, the degree of smoothing is not an independent validation metric; it only evaluates the ability of reconstruction techniques to transfer input data to the output (Barth et al., 2022). This analysis was performed on a logarithmic scale, consistent with the input and output of the model for performance evaluation.





### 2.4.3 Independent validation using in situ measurement

For validating satellite products with in situ data, the protocol from Bailey and Werdell (2006) and the guidelines from the EUMETSAT Sentinel-3 OLCI Ocean Colour product Matchup Protocols (EUMETSAT, 2022) are followed. The key steps include:

- Pixels are matched based on the in situ data points within the 3×3 pixel box and captured on the same day.
- A minimum of 50 % +1 of the 'valid pixels' within a 3×3 pixel box (i.e., at least 5 pixels) is required to retain the matchup.
- Pixels with deviations exceeding ±1.5 times the standard deviation are removed as outliers.
- Matchups are discarded if the coefficient of variation of the remaining pixels exceeds 0.2.

The comparison between the satellite product and in situ measurements is conducted by evaluating the coefficient of determination ($R^2$), slope, and intercept of the regressions, based on logarithmically scaled satellite ($log_{10}(C_{st})$) versus logarithmically scaled in situ measurement ($log_{10}(C_{in})$). Additionally, the median-percentage-deviation (MedPD), root-mean-square-deviation (RMSD), and normalized RMSD (NRMSD) (or relative RMSD) are calculated using linear data. The model performance statistics are presented as follows:

$$log_{10}(C_{st,i}) = Intercept + Slope \times log_{10}(C_{in,i}) + \epsilon \tag{7}$$

$$R^2 = \frac{\sum_{i=1}^{M}(log_{10}(C_{st,i}) - log_{10}(\overline{C_{in}}))^2}{\sum_{i=1}^{M}(log_{10}(C_{in,i}) - log_{10}(\overline{C_{in}}))^2} \tag{8}$$

$$MedPD = Med\left(\frac{|C_{st,i} - C_{in,i}|}{C_{in,i}} \times 100\ \%\right) \tag{9}$$

$$RMSD = \sqrt{\frac{1}{M}\sum_{i=1}^{M}(C_{st,i} - C_{in,i})^2} \tag{10}$$

$$NRMSD = \frac{RMSD}{\overline{C_{in}}} \tag{11}$$

Model II regression is used when both the response and explanatory variables are prone to measurement errors, allowing for more accurate comparisons by accounting for uncertainties in both variables (Legendre, 1998; Legendre and Legendre, 2012). This study applied model II regression (major axis) to account for the uncertainty in DPA-derived PFTs when validating satellite products. This method was not used for TChla, as it was derived using a more direct approach with lower associated uncertainty. The uncertainty associated with pigment-based Chla measurements is approximately 7 % for TChla and higher for other groups (Claustre et al., 2004; IOCCG, 2019).





## 3 Results and Discussion

### 3.1 Hyperparameter tuning and final model configuration

During model development, the validation dataset is used to tune hyperparameters and determine the optimal model configuration. The tuning process focuses on optimising the model performance and ensuring robust generalisation. Following the methodology outlined in Sect. 2.3.3, we conducted hyperparameter tuning to assess the performance of both gap-filling methods in area No. 9. A random search strategy was employed for hyperparameter exploration, with 20 permutations for DINEOF and over 100 permutations for DINCAE. The number of hyperparameters in DINCAE significantly exceeds those in DINEOF, requiring more permutations for optimal tuning (Table S1). DINEOF's model configuration results on the validation dataset show that the total RMSLE remains highly consistent across all experiments, averaging $0.122 \pm 0.001$ $\log_{10}(\text{mg} \cdot \text{m}^{-3})$, suggesting that further optimization would yield minimal improvements. In contrast, DINCAE exhibits greater variability in performance, with some trials failing to converge and getting stuck in local minima, resulting in unrealistic RMSLE values even exceeding $10^3$ (around 30 % of experiments). Even when DINCAE successfully converges to an optimal minimum (RMSLE < 0.3) in 25 % of experiments, its performance remains more variable with an average of $0.176 \pm 0.044$ $\log_{10}(\text{mg} \cdot \text{m}^{-3})$. Despite this variability, both models achieve similar minimum total RMSLE on the validation dataset ($0.12$ $\log_{10}(\text{mg} \cdot \text{m}^{-3})$), indicating comparable fits to the validation dataset. Notably, the optimal configured model for both gap-filling methods does not require SST as an auxiliary dataset, instead relying solely on Chla datasets for gap-filling, consistent with the findings of Han et al. (2020) for TChla gap-filling. The final optimal hyperparameter settings for the two models are presented in Table S2 and Table S2.

Once the optimal model configurations for both gap-filling methods are determined, the combined training and validation datasets are used to reconstruct the data across all areas. Figure 4b presents the number of EOFs extracted for each area using the DINEOF method. Areas close to continental shelves and coastal regions, such as areas No. 1, 2, 9, and 10, generally require more EOFs to capture their variability. In contrast, open ocean and oligotrophic areas like area 6 need fewer EOFs to describe their variability. However, areas with a high rate of missing data, such as area No. 7, face challenges in pattern extraction, leading to fewer EOFs extraction. The reconstructed data for each gap-filling model across different areas are combined using a weighted blending method to ensure a smooth transition between regions. An example layout of the fully reconstructed data for DINEOF and DINCAE outputs is shown in Figure 4. As illustrated in the original satellite data (Figure 4a), a significant portion of the data is missing, and a comparison with the reconstructed outputs from both models highlights their substantial ability to reconstruct data from limited observations. The visual comparison of the two gap-filling outputs for diatoms on 26 May 2018 reveals noticeable differences in the reconstructed patterns, particularly in areas No. 1, 7, and 10, where distinct bloom patterns are observed. These differences highlight the need for further investigation through evaluation techniques.



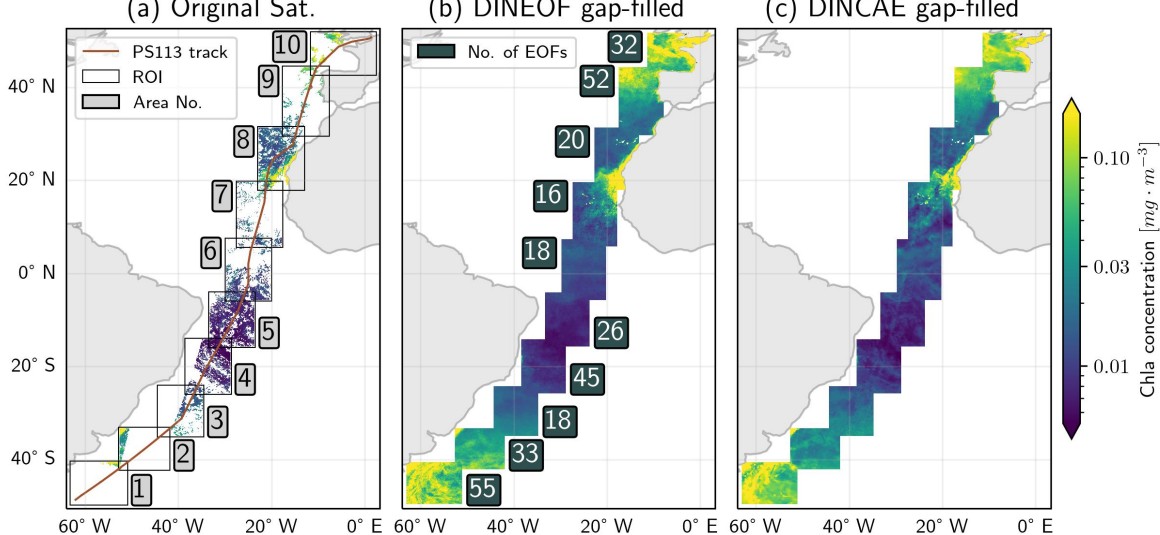

**Figure 4: Example of input and final output of gap-filling methods for diatoms on 26 May 2018. (a) Original satellite product fed to the gap-filling models, with squares indicating different areas (ROI) used for gap-filling. The red line represents the locations of the research vessel throughout the PS113 expedition. (b) Merged output of the DINEOF gap-filling method. The values within the blue boxes indicate the number of EOFs extracted during the DINEOF gap-filling process. (c) Merged output of the DINCAE gap-filling method.**

## 3.2 Performance evaluation

During the data partitioning process, a subset of the available data was masked to serve as a test dataset for evaluating gap-filling techniques and conducting performance comparisons (explained in Sect. 2.2.3). The spatial variation in the average absolute logarithmic difference between the reconstructed data and the test dataset is illustrated in Figure 5. Errors are observed to exceed 0.3 $\log_{10}(\text{mg·m}^{-3})$ in certain regions, particularly along the West African coast (areas No. 7 and 8), the Argentine Sea (areas No. 1 and 2), and the English Channel (area No. 10) with high phytoplankton abundance. Significantly lower error values are recorded in the oligotrophic regions of the South Atlantic gyres with lower phytoplankton abundance. The error distributions are similar between TChla and all PFTs. However, the errors for prokaryotes appear more consistent across regions, likely due to their relatively stable Chla concentration of prokaryotes. The errors for TChla are lower, as greater data availability and higher concentration values facilitate more accurate gap-filling. In contrast, diatoms and green algae show high errors, even in open ocean and oligotrophic regions. The error difference analysis (The last row of Figure 5) reveals that the DINCAE model consistently outperforms the DINEOF model in reconstructing TChla and PFTs, exhibiting lower absolute differences, particularly in hotspots of high errors (e.g., coastal areas, continental shelves, and equatorial regions) where DINEOF demonstrates notable discrepancies. The performance difference between the two gap-filling methods can exceed ±0.2 $\log_{10}(\text{mg·m}^{-3})$ across all groups. However, the average error difference is relatively small, as the majority of differences in the open ocean are really small. The average error differences are 0.03 for TChla, 0.01 for diatoms, 0.01 for dinoflagellates, 0.02 for haptophytes, 0.02 for green algae, and 0.01 $\log_{10}(\text{mg·m}^{-3})$ for prokaryotes.





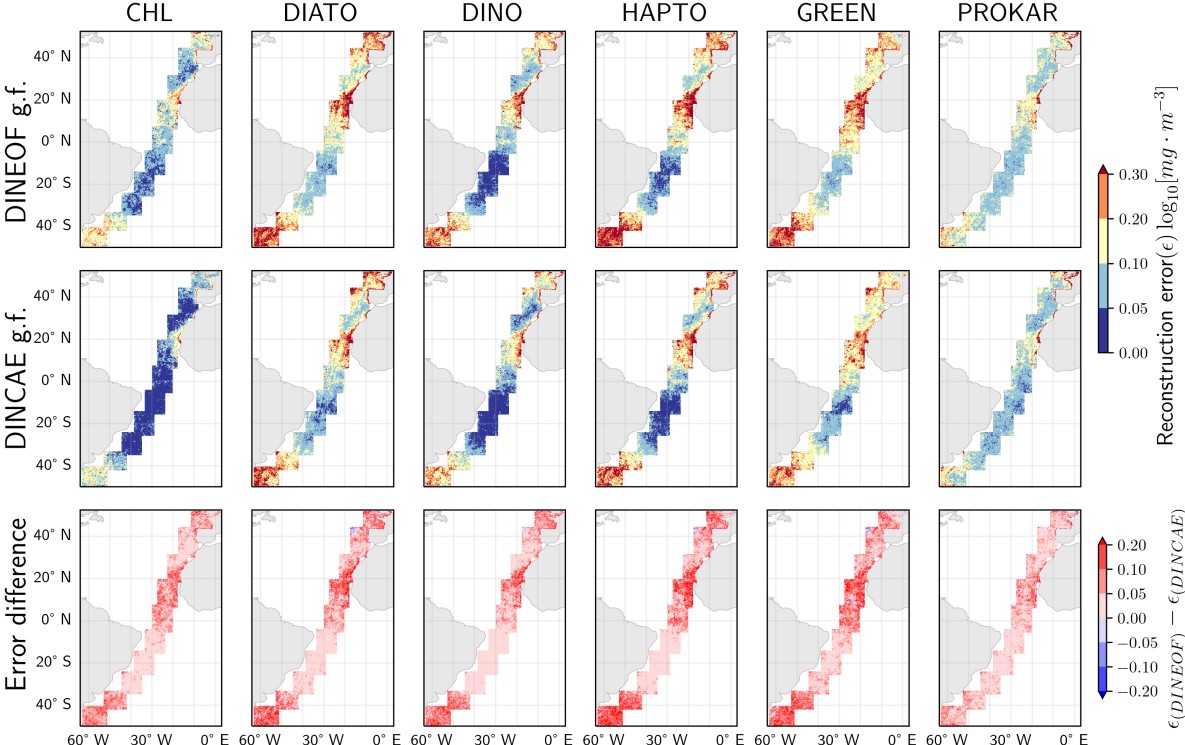

**Figure 5: Spatial variation in the average absolute logarithmic differences between the gap-filled (g.f.) and the test dataset, along with a comparative analysis of the two gap-filling models.**

The RMSLE and MAPE between the DINEOF and DINCAE reconstructed data and the test dataset across all areas and phytoplankton groups are presented in Figure 6. The top panel of Figure 6 shows that DINCAE's RMSLE for TChla ranges from 0.03 to 0.12 $\log_{10}$(mg·m$^{-3}$), from the oligotrophic region to the high productivity zones near the Patagonian shelf and the English Channel, while DINEOF's RMSLE is slightly higher, ranging from 0.05 to 0.16 $\log_{10}$(mg·m$^{-3}$). The PFTs show marginally higher errors, with RMSLEs between 0.03 and 0.25 $\log_{10}$(mg·m$^{-3}$) for DINCAE reconstruction, and between 0.04 and 0.30 $\log_{10}$(mg·m$^{-3}$) for DINEOF reconstruction. Similar to TChla, the highest RMSLE for PFTs in both models is observed on the Patagonian shelf and near the English Channel, while the lowest RMSLE occurs in the oligotrophic region, where phytoplankton abundance is low. On average, the RMSLE of DINEOF is observed to be approximately 66 % higher for TChla and 16 % higher for PFTs compared to DINCAE (11 % for diatoms, 20 % for dinoflagellates, 22 % for haptophytes, 16 % for green algae, and 12 % for prokaryotes). The findings suggest that DINCAE generally exhibits a lower RMSLE than DINEOF across most regions and phytoplankton groups.

To enable a unit-free comparison across PFT groups and regions, statistical analysis is performed using MAPE (Figure 6, bottom panel). MAPE values are generally lower for TChla than that for PFTs. The averaged MAPE obtained by DINEOF and DINCAE reconstructions in coastal regions and continental shelves (area No. 1, 2, 7, 8, 9, 10) are approximately 32 % and 26 %, respectively, while in the open ocean (area No. 3, 4, 5, 6), the errors are averaged to 13 % and 10 %, respectively.



A significant reduction in errors is achieved with DINCAE reconstruction compared to DINEOF, with notable improvements being observed for PFTs, particularly diatoms, dinoflagellates, and haptophytes in areas No. 1 and 7. MAPE is found to be notably higher in regions with increased dynamics due to coastal activity (areas No. 1, 2, 8, 9, and 10) or in areas with a high
rate of missing data (area No. 7), while lower MAPE values (approximately 10 %) are observed in areas No. 3, 4, 5, and 6, characterized by lower dynamics and phytoplankton abundance in oligotrophic zones. The highest MAPE is recorded in area No. 1 for haptophytes, with 67 % and 48 %, for diatoms with 57 % and 43 %, and for green algae with 53 % and 43 %, for DINEOF and DINCAE reconstructions, respectively. Dinoflagellates and prokaryotes exhibit comparatively smaller errors. Even in unit-free comparisons, prokaryotes, despite their low concentration, maintain a relatively consistent MAPE ranging
from 10 % to 22 % across all areas.

Our results show notable similarities and differences compared to previous studies. Sirjacobs et al. (2011) gap-filled four years of daily TChla (from MERIS), TSM, and SST data for the Southern North Sea and English Channel. They reported RMSLE values for TChla ranging from 0.09 to 0.29 $\log_{10}(\text{mg} \cdot \text{m}^{-3})$. Hilborn and Costa (2018) gap-filled three years of MODIS-Aqua TChla dataset for the highly productive coastal region of the Salish Sea, reporting an RMSLE ranging from
0.17 to 0.22 $\log_{10}(\text{mg} \cdot \text{m}^{-3})$ for daily data and 0.27 to 0.32 $\log_{10}(\text{mg} \cdot \text{m}^{-3})$ for weekly composite data. Wang et al. (2019) reported an RMSLE of 0.13 $\log_{10}(\text{mg} \cdot \text{m}^{-3})$ during the development of a long-term cloud-free Chla dataset derived from SeaWiFS and MODIS satellite observations over the Bohai and Yellow Seas. Han et al. (2020) reconstructed TChla in the South China Sea and West Philippine Sea using the daily merged (GlobColour) product with DINEOF and DINCAE gap-filling techniques. Their results reported a minimum cross-validation error of 0.11 $\log_{10}(\text{mg} \cdot \text{m}^{-3})$ (converted from
$\ln(\text{mg} \cdot \text{m}^{-3})$) for DINCAE and 0.12 $\log_{10}(\text{mg} \cdot \text{m}^{-3})$ for DINEOF in the South China Sea, and 0.12 $\log_{10}(\text{mg} \cdot \text{m}^{-3})$ for DINCAE and 0.13 $\log_{10}(\text{mg} \cdot \text{m}^{-3})$ for DINEOF in the West Philippine Sea. In comparison, the RMSLE in our TChla reconstruction is notably lower, with values ranging from 0.08 to 0.16 $\log_{10}(\text{mg} \cdot \text{m}^{-3})$ for DINEOF and 0.03 to 0.12 $\log_{10}(\text{mg} \cdot \text{m}^{-3})$ for DINCAE reconstructions. The higher errors in their study likely reflect the high Chla concentrations of TChla and the more dynamic nature of their study region. In our study, the RMSLE values for PFTs are comparable to those reported for TChla
in literature, with the highest values observed for diatoms, ranging from 0.07 to 0.28 $\log_{10}(\text{mg} \cdot \text{m}^{-3})$ for DINEOF and 0.06 to 0.25 $\log_{10}(\text{mg} \cdot \text{m}^{-3})$ for DINCAE reconstructions. Ji et al. (2021) used DINCAE for the gap-filling of SST and TChla in the East China Sea using MODIS-Aqua, MODIS-Terra, and VIIRS-SNAP products. Their cross-validation results reported a mean relative error (MRE) of 0.40, corresponding to a minimum MAPE of 40%. In comparison, the MAPE for TChla and prokaryotes in our study is consistently below 40% across all regions, and for the remaining PFTs, it is generally below or
comparable to this value in productive regions. These results indicate that the validation errors for PFT reconstructions for both methods fall within an acceptable range relative to those reported in the literature.

In summary, both methodologies demonstrate substantial capability in reconstructing TChla and PFTs with minimal error in open ocean settings, while maintaining acceptable relative error in coastal and high-dynamic regions. The highest MAPE are 67 % for DINEOF and 48 % for DINCAE, lower than the uncertainty levels of the original PFT products (average
uncertainty for the study period and the entire Atlantic Ocean as defined in Sect. 2.1.2: TChla 33 %, diatoms 140 %,



dinoflagellates 112 %, haptophytes 122 %, green algae 88 %, and prokaryotes 102 %). The DINCAE method consistently exhibits superior performance, significantly reducing errors in comparison to DINEOF. These results highlight its robustness across all regions and phytoplankton groups, even in the face of challenges posed by the high temporal and spatial dynamics involved in reconstructing data in coastal areas (e.g., areas No. 1, 7, 9, and 10), demonstrating its ability to distinguish and

reproduce patterns within the dataset based on a limited amount of available data. However, it is important to note that this improved accuracy comes at the cost of more intensive computational demands, requiring GPU resources and a higher number of tuning permutations.

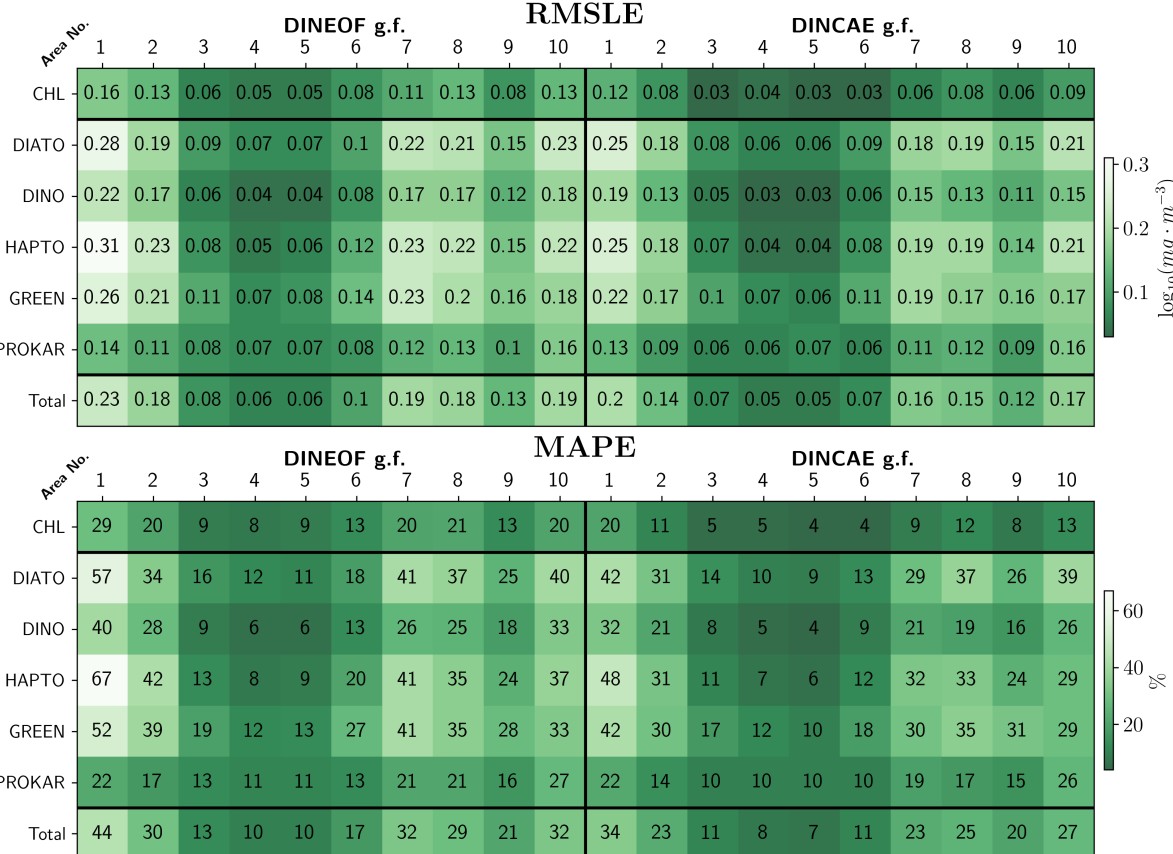

**Figure 6: Statistical outcome and comparative analysis for the two gap-filling (g.f.) models on the test dataset**

**3.3 Spatial smoothing**

**3.3.1 Gradient field**

As detailed in the Sect. 2.4.2, the Sobel Edge detection algorithm is used to compute the gradient field in the TChla and PFTs concentration. As an example, results for one test dataset date in area No. 10 are presented in Figure 7. In the Celtic Sea (Figure 7 box 1), a pronounced gradient field is evident in the original satellite data for TChla and diatoms, which is





removed when clouds are added during test dataset generation. Both algorithms successfully reconstructed the high gradient in TChla, maintaining a similar magnitude and pattern. DINCAE produced a gradient pattern closer to the large-scale features of the original satellite data, while DINEOF better captured smaller-scale patterns. For diatoms, DINCAE produced a gradient pattern more consistent with the original satellite product compared to DINEOF. In the Bay of Biscay, along the western coast of France (Figure 7 box 2), a significant gradient change is observed in the TChla dataset, while the diatom

dataset was initially missing. After reconstruction, both DINEOF and DINCAE transferred the gradient pattern from TChla to diatoms, demonstrating their ability to capture relationships between datasets. However, DINCAE produced a diatom gradient pattern more closely aligned with TChla than DINEOF. Overall, the DINEOF gradient appeared noisier than the original satellite and DINCAE gap-filled products. Additionally, satellite track patterns are noticeable in DINEOF diatom reconstructions but are lower in DINCAE outputs.

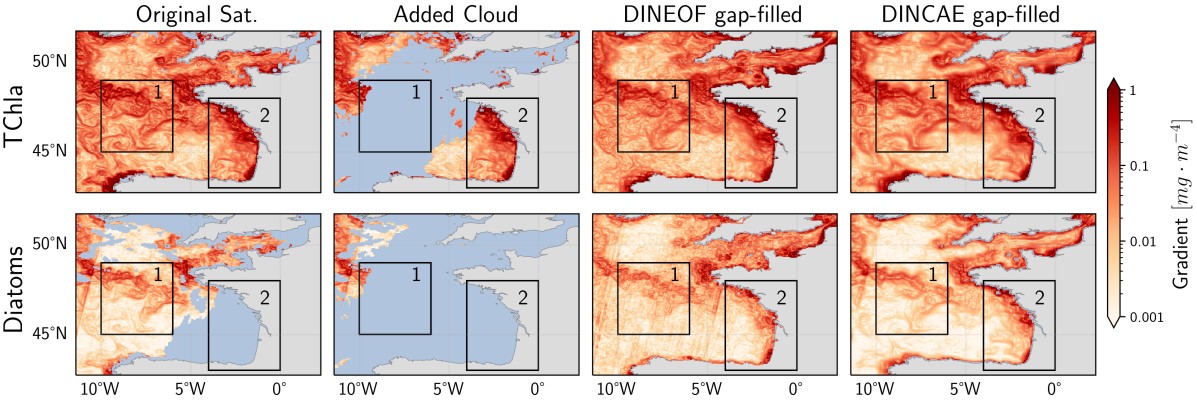


**Figure 7: Gradient Field [mg·m⁻⁴] of TChla and diatoms for original Satellite data, with added cloud, DINEOF reconstructed, and DINCAE reconstructed datasets computed for the 23 June 2018 on area No. 10. The blue colour shows the missing values.**

### 3.3.2 Degree of smoothing of originally present data

Figure 8 compares the degree of smoothing as RMSLE between the original satellite product against the DINEOF and

DINCAE reconstructed datasets for TChla and PFTs. The lowest RMSLE are observed for TChla, prokaryotes, and dinoflagellates, with around 0.10 and 0.05 $\log_{10}$(mg·m⁻³) for DINEOF and DINCAE, respectively. The highest RMSLE are associated with diatoms and green algae in both datasets, with around 0.14 and 0.07 for DINEOF and DINCAE reconstruction, respectively. The results show that DINEOF has approximately twice the RMSLE of DINCAE for all groups. This indicates that the DINCAE algorithm more effectively transfers the available data from the original satellite dataset to

the reconstructed dataset with lower deviation and less smoothing.



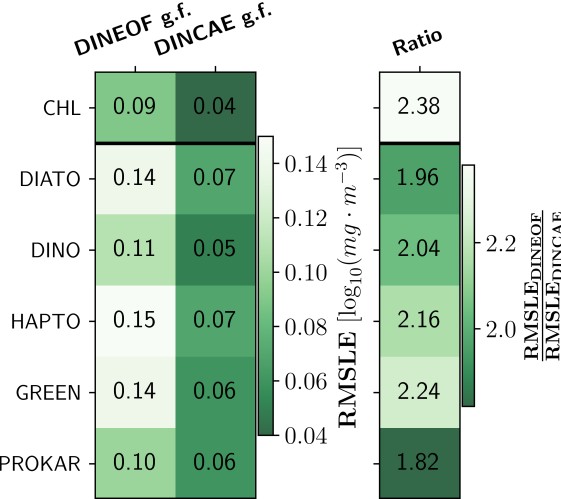

**Figure 8: Comparison of the degree of smoothing between the two gap-filling (g.f.) methods for TChla and PFTs. The ratio indicates the degree of smoothing achieved by DINEOF relative to DINCAE reconstruction.**

### 3.4 Independent validation

Figure 9 and Figure 10 evaluate the matchups between the original satellite, gap-filled DINEOF or gap-filled DINCAE products against in situ measurements. The evaluation of DINEOF and DINCAE reconstructed data in comparison with in situ measurements was conducted by categorizing the dataset into two groups: transferred matchups, representing matchups present in the original satellite product, and filled matchups, referring to the pixels that were missing in the original satellite product and filled through the reconstruction method. In Figure 9, the regression analysis for both methods across these

categories is illustrated, and Figure 10 provides a summary of the statistical parameters for performance comparison, distinguishing between transferred and filled indices. The transferred matchup points used for this validation are consistent with those used for validating the original satellite product, enabling a direct comparison to assess whether the accuracy of the original satellite data is preserved in the reconstruction process. The first columns in Figure 9 and the original satellite statistical description in Figure 10 demonstrate that the TChla product from the original satellite dataset exhibits superior

validation results compared to PFTs. This is attributed to the lower uncertainty associated with the TChla product, which is derived from maturer conventional ocean colour algorithms that have reduced the product uncertainty as low as ~30% on average, whereas PFT concentrations, which are derived more indirectly from pigment concentrations via DPA as ground truth data, posing greater challenges for accurate interpretation and separation from the total biomass by the backscattered signals measured by the satellite sensors. TChla obtains more matchups, due to higher satellite data availability due to the

usage of multiple sensor data, with most points aligning closely along the 1:1 line. In addition, the regression analysis shows that TChla has more favourable slope (0.78) and intercept (-0.38) values compared to the PFTs, which have slopes ranging from 0.41 to 0.66 and intercepts from -0.28 to -1.11. The coefficient of determination ($R^2$) is approximately 0.84 for TChla and diatoms, around 0.69 for haptophytes and green algae, and about 0.5 for dinoflagellates and prokaryotes. The MedPD of





35% and deviations below the 1:1 line at high Chla concentrations indicate an underestimation of TChla by the original
satellite product. In contrast, for PFT products, this trend is primarily associated with overestimations at low Chla
concentrations. The highest MedPD values are observed for diatoms and dinoflagellates, at 121% and 98%, respectively. The
remaining PFTs exhibit MedPD values comparable to TChla, approximately 35%. The highest RMSDs are observed for
TChla and diatoms, around 0.23 mg·m⁻³, while other groups have lower RMSDs of approximately 0.03 mg·m⁻³. However,
because RMSDs vary with concentration ranges, they were normalized to NRMSD for better comparison. The NRMSD
reveals the lowest error for prokaryotes, at 0.51, and the highest for diatoms, with an NRMSD of 4.66.

The transferred matchup validation results indicate minimal variation in statistical outcomes when compared to the original
satellite validation, with minor deviations observed in R² and MedPD values before and after reconstruction. Notably, the R²
values for TChla, diatoms, dinoflagellates and green algae in the DINEOF reconstruction, along with the MedPD for diatoms
in both methods, surpass those of the original satellite data. Overall, the differences in other statistical metrics are negligible,
suggesting that both DINEOF and DINCAE effectively preserve the original patterns of the input dataset in the reconstructed
outputs. Upon initial observation of Figure 9, it is evident that the filled matchups exhibit greater dispersion compared to the
transferred matchups. As expected, the number of matchups generated through gap-filling in satellite products substantially
exceeds those found in the original satellite data. The total number of matchups for the fully reconstructed dataset is
approximately 2.35 times greater for TChla and 5 to 6 times greater for PFTs than the number of original satellite matchups.
Figure 10 demonstrates that, for both methods, the slopes and intercepts of the filled matchups are slightly lower than those
of the transferred matchups for TChla and prokaryotes and nearly identical for diatoms. Furthermore, the filled matchups
exhibit improved performance for dinoflagellates, haptophytes, and green algae. A lower R² is observed in all filled
matchups of the DINCAE and DINEOF reconstructions compared to the transferred matchups, indicating significant
dispersion from the regression line. MedPD and RMSD of filled matchups indicate reduced accuracy in most cases
compared to the transferred matchups, except diatoms and dinoflagellates for both methods. The NRMSD reveals greater
error levels for TChla, haptophytes, green algae, and prokaryotes, while diatoms and dinoflagellates exhibit lower errors.
Notably, the RMSD for TChla in filled matchups is approximately 2.75 times that of transferred matchups, whereas the
NRMSD is only about 1.12 times higher, a pattern consistent across all PFTs. The higher overall mean Chla concentration
for the filled matchups could contribute to the increased RMSD values as compared to the transferred. These findings
suggest that, in most instances, the accuracy of the gap-filled datasets is slightly worse than the original satellite data.
Nevertheless, the filled matchups generally follow the same trends and trajectories, except for prokaryotes, albeit with
increased variability relative to the transferred matchups. The ability of satellite gap-filling methods to preserve accuracy in
existing matchups while significantly expanding the number of matchups with acceptable accuracy highlights the efficiency
of these techniques.

The transferred and filled matchups are combined to facilitate a comparison of the outcomes of the reconstruction methods
and to evaluate their respective accuracy. Figure 10 presents the statistical results of these regression analyses. The slope and
intercept of the regressions are generally better for the DINEOF gap-filled product compared to the DINCAE gap-filled



product and even the original satellite product. Exceptions are observed for green algae, where the slope is slightly lower, and for prokaryotes, where it is substantially lower for both gap-filling methods compared to the original dataset. This
reduction in accuracy arises from the gap-filling process and reflects the distinct abundance patterns of prokaryotes compared to other groups. Future studies should consider separating the gap-filling process for prokaryotes to improve the performance across all groups. For TChla, the $R^2$ values for both gap-filled products are similar to those of the original satellite product, with DINEOF slightly outperforming. For PFTs, however, the $R^2$ values are significantly better for DINEOF compared to DINCAE, with DINEOF achieving results closer to the original satellite product. however, the $R^2$
values are significantly better (enhanced by 27% - 117% for different PFTs) for DINEOF compared to DINCAE, with DINEOF achieving results closer to the original satellite product. MedPD for the gap-filled products is approximately 34% lower for diatoms and dinoflagellates, 60% lower for green algae, and 22% lower for prokaryotes compared to the original satellite dataset. RMSD values for the gap-filled products are also close to each other; however, they are twice as high as the original satellite product for TChla and diatoms, and five times higher for green algae. Normalizing to NRMSD reveals that
these differences are primarily due to the average Chla concentrations in the filled products. The NRMSD of gap-filled TChla is only about 17% higher than the original product, while it is 17% lower for diatoms and 30% lower for green algae. Overall, for TChla, both gap-filling methods demonstrate robustness comparable to the original satellite dataset validation, albeit with higher RMSD, while maintaining similar NRMSD. For PFTs, DINEOF generally outperforms DINCAE, particularly in slope, intercept, and $R^2$ metrics. These results highlight the superior performance of DINEOF in external
validation.

Xi et al. (2021) validated the global merged OC satellite product for TChla and PFTs, reporting metrics in the order (MedPD, RMSD [mg·m⁻³], $R^2$) as follows: TChla (32 %, 1.08, 0.82), diatoms (56 %, 0.92, 0.77), dinoflagellates (54 %, 0.89, 0.62), haptophytes (43 %, 0.16, 0.71), green algae (52 %, 0.10, 0.53), and prokaryotes (42 %, 0.09, 0.46). Comparisons with the original satellite validation in this study show that the MedPD ranges are similar, except for diatoms and dinoflagellates,
which are approximately double those reported by Xi et al. Notably, their MedPD values align closely with the gap-filled MedPD results in this study. However, their RMSD values are at least twice as high for TChla and all PFTs, exceeding both the original satellite and gap-filled product results. This is primarily because their focus was on a global scale, encompassing greater variability compared to our study area. Regarding $R^2$, Xi et al. (2021)'s values are comparable to ours for TChla and haptophytes, lower for diatoms, green algae, and prokaryotes but higher for dinoflagellates. Xi et al. (2023b) used data from
16 expeditions across the Atlantic Ocean to validate long-term trends in four PFT Chla monthly products: diatoms, haptophytes, prokaryotes, and dinoflagellates. However, due to missing satellite data and inconsistencies during matchup extraction, less than 10 % (192 out of 1975) of the in situ measurements could be used as matchups for validating the satellite-derived PFT products, similar to the number of matchups in our study. A comparison of the matchup statistical results from Xi et al. (2023b), involving the monthly satellite products and in situ measurements, with the statistical results
from our gap-filled products reveals slightly different performance: Their comparison showed stronger performance in terms of slope (diatoms 0.71, haptophytes 0.95, prokaryotes 0.71, dinoflagellates 1.07) and intercept (diatoms -0.27, haptophytes -



0.01, prokaryotes 0.12, dinoflagellates 0.04). Results for $R^2$ (diatoms 0.76, haptophytes 0.41, prokaryotes 0.36, dinoflagellates 0.66), MedPD (diatoms 60 %, haptophytes 59 %, prokaryotes 185 %, dinoflagellates 59 %), and RMSD (diatoms 0.30, haptophytes 0.18, prokaryotes 0.06, dinoflagellates 0.07) were mixed, with some cases favouring their study and others DINEOF matchups, appearing overall comparable. DINCAE performed worse than their matchups in terms of slope, intercept, and $R^2$. For MedPD and RMSD, the results were mixed, similar to the trends observed with DINEOF.

Previous research employing the DINEOF and DINCAE gap-filling techniques has predominantly focused on SST, TChla, and SPM, with no prior studies addressing PFTs. Consequently, our validation results can only be compared with previous findings for TChla, although TChla is not the primary focus of our investigation. Alvera-Azcárate et al. (2021) reconstructed 23 years of Chla and SPM data in the Greater North Sea using the DINEOF method. Their validation results, based on regression analysis of matchups between the original satellite and DINEOF reconstructed products with in situ measurements, demonstrated a decline in $R^2$ from 0.75 to 0.58 after reconstruction, with the MAE increasing from 2.47 to 2.83 mg·m$^{-3}$. Our study indicates a smaller decrease in $R^2$ from 0.84 to 0.82 and an increase in MAE from 0.10 to 0.19 mg·m$^{-3}$ using the DINEOF method for TChla matchups. We observe reduced scatter compared to the previous study, despite a higher reconstruction error in our results. However, the magnitude of the MAE differs significantly between the two studies, largely due to an overall higher phytoplankton abundance in the North Sea compared to the Atlantic Ocean. In another study, Barth et al. (2021) reconstructed 20 years of TChla and SPM data in the southern North Sea using DINCAE. Their validation using the original satellite and DINCAE reconstructed data with in situ measurement, indicated a decline in the validation slope from 0.82 to 0.64, an increase in the intercept from 0.07 to 0.33, a reduction in $R^2$ from 0.62 to 0.41, an increase in log10-RMSD from 0.29 to 0.33, and an increase in number of matchups from 25 to 27. Our application of DINCAE to TChla results in a similar slope of 0.78, an intercept shift from -0.38 to -0.35, a reduction in $R^2$ from 0.84 to 0.80, an increase in log10-RMSD from 0.24 to 0.25, and an increase in matchups from 94 to 221. Volpe et al. (2018) developed an operational gap-filling technique based on DINEOF for ocean colour products in the Mediterranean Sea, validated using 1643 in situ measurements collected between 1997 and 2015. Their results reported RMSE values of 0.27 for the Level 3 product (original satellite dataset) and 0.29 for the Level 4 product (gap-filled), with absolute percentage difference (APD) values of 56 % for Level 3 and 53 % for Level 4 during the operational phase. In comparison, the external validation results for TChla in our study show an RMSE of 0.25 for Level 3, 0.52 for DINEOF Level 4, and DINCAE Level 4 products, with a MedPD of -35 % for Level 3, -33 % for DINEOF Level 4, and -36 % for DINCAE Level 4. The findings of Volpe et al. (2018) indicate an increase of approximately 6.6 % in RMSE and a 3 % decrease in APD for gap-filled products. In our study, however, RMSE for gap-filled TChla products increased by 108 % for both methods, while MedPD (noting that APD and MedPD may not be directly comparable) changed by -1 % for DINEOF and 1 % for DINCAE gap-filling. These results suggest that in our study, the median validation error (MedPD) is nearly identical between the original satellite and gap-filled TChla products, though higher RMSD indicates more extreme errors at both high and low concentrations. For PFTs, the most significant improvements in MedPD are observed for diatoms (-39 % for DINEOF and -41 % for DINCAE gap-filled products) and dinoflagellates (-38 % for DINEOF and -33 % for DINCAE). The remaining





PFTs and TChla exhibit marginal reductions or values similar to the original satellite MedPD, comparable to the APD values for TChla reported in Volpe et al. (2018). RMSD for dinoflagellates and prokaryotes shows slight changes, consistent with the TChla level in Volpe et al. (2018), whereas other PFTs and TChla display more substantial increases in RMSD. A comparison of the external validation results from the literature with those from our study indicates that the errors are

generally lower and within an acceptable range, demonstrating the robustness of both the gap-filled datasets and the external validation.

In summary, DINEOF slightly outperforms DINCAE in validation against in situ measurements. In regression analyses (slope, intercept, and $R^2$), DINEOF demonstrates superior performance relative to DINCAE, occasionally even surpassing the original satellite products. This is particularly evident in the case of diatoms, dinoflagellates, haptophytes, and green

algae for slope and intercept, and dinoflagellates for $R^2$.









**Figure 9: Regression analysis comparing in situ measurements with the original satellite product, DINEOF, or DINCAE reconstructed data. The matchups for the reconstructed data are also categorised into two additional groups: transferred matchups (representing data points present in the original satellite product) and filled matchups (representing data points missing**

**in the original satellite product and subsequently filled by the reconstruction models).** *n* **refers to the number of matchups between the satellite products and the in situ measurement.**

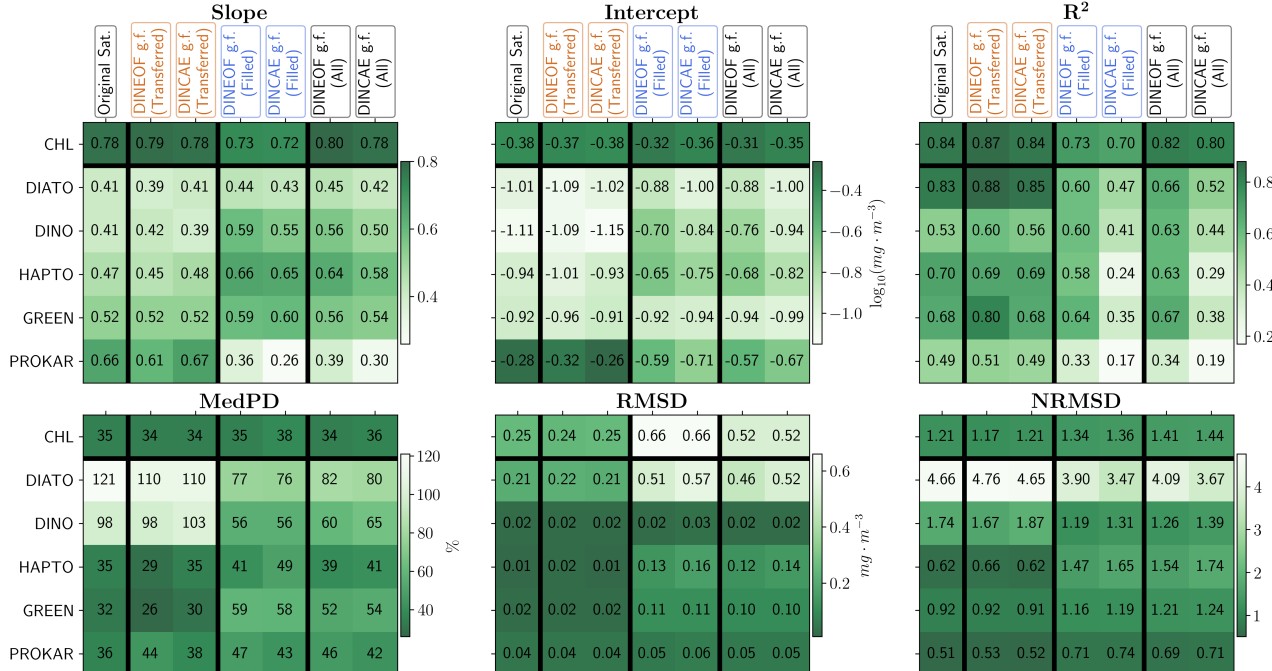

**Figure 10: Statistical outcome of the regression analysis comparing in situ measurements with the original satellite product, DINEOF, or DINCAE reconstructed data based on filled, transferred and all matchups.**

**3.5 Novelty and limitations**

A comprehensive understanding of phytoplankton composition and distribution is crucial for explaining biogeochemical processes and assessing the impact of climate change on marine ecosystems and biodiversity. However, the methods available for retrieving phytoplankton dynamics and distributions are currently constrained. In situ measurements are limited in spatiotemporal coverage and fail to represent the full extent of phytoplankton dynamics. Most current biogeochemical

models are limited in the diversity of phytoplankton groups, often focusing on diatoms (micro-phytoplankton) and prokaryotes (pico-phytoplankton) (e.g., RECOM (Schourup-Kristensen et al., 2014) and PISCES (Aumont et al., 2015)) and assimilation of satellite-derived TChla and PFT products shown to be effective in improving the models' performance in predicting these phytoplankton groups (e.g., Pradhan et al., 2019, 2020). Additionally, these models require evaluation by observations with high temporal and spatial coverage and provided uncertainty. Satellite observations of TChla and PFT

Chla have covered the global ocean for more than 20 years. However, these data are significantly limited in coverage due to non-optimal observing conditions and sensors' availability in operation, leading to data gaps exceeding 90% in some critical



regions. So far, operational gap-filled PFT Chla products are unavailable, which may have been due to the especially high PFT product data gap rates and the substantial computational demands of gap-filling techniques.

In this study, we applied for the first time gap-filling methods to PFT Chla products. We used two well-established gap-filling techniques, DINEOF and DINCAE, that have proven effective in reconstructing ocean satellite data products, specifically SST and TChla. We subsequently evaluated the performance of the two methods using multiple techniques. This novel application showed new perspectives on the gap-filling of multivariate datasets and their validation, particularly for phytoplankton community structure. We showed the transfer of patterns from broader parameter categories, like TChla, to more specific subcategories, such as PFTs. Both gap-filling methods demonstrated robustness, even in areas with high data missing rates. Notably, gap-filling of PFT products resulted in approximately 80 % more data with minimal impact on the original satellite data, enhancing the understanding of biogeochemical dynamics at the spatiotemporal scale (e.g., time-series analysis) and increasing the number of matchups with in situ measurements by a factor of 5 to 6, thereby supporting further model development and validation. The reconstructions demonstrated significant efficiency in capturing transient-scale oceanic features, with DINEOF using more EOFs to retain various patterns, while DINCAE employing advanced machine learning techniques, such as skip-connections, to effectively preserve these features. For these developments, we used datasets from various Longhurst biogeochemical provinces in the Atlantic Ocean, ensuring the methods' applicability across diverse oceanic conditions, ranging from oligotrophic to eutrophic regions and from open ocean to continental shelves and coastal areas. The successful reconstructions of data across different oceanic regimes underscores the potential for these methods to be extended to other oceanographic variables and regions, paving the way for improved environmental monitoring and predictive modelling efforts on a global scale.

While both models demonstrated remarkable proficiency in reconstructing TChla and PFTs, certain limitations remain that offer opportunities for further improvement in future work. Both approaches face inherent limitations in scaling a single scene analysis spatially and are specifically designed for regional applications involving multivariate long-term analysis. The associated computational demands further constrain their feasibility for broader, global-scale implementation. Consequently, multiple areas had to be reconstructed along the corridor of the expedition to obtain sufficient matchups for consistent validation purposes. The in situ measurements are limited to the expedition duration, which may affect the robustness of the external validation process for assessing the gap-filling model over several years of data in the Atlantic Ocean. Incorporating datasets from other expeditions conducted at different times in the same region, as implemented by Xi et al. (2023b), enhances the robustness of validation by extending the temporal and spatial coverage of in situ measurements, thereby reducing potential validation biases. Furthermore, both models incorporate hyperparameters that require multiple random search permutations to achieve a robust architecture, like most machine learning algorithms. This requirement is particularly pronounced for DINCAE, which features a convolutional structure, compared to DINEOF, which is inherently parameter-free. Even with optimised structures selected for the development area, the spatial transferability of the hyperparameter combinations needs to be tested.





Moreover, neither model includes per-pixel uncertainty of the input dataset, which is crucial for more accurate reconstruction. DINEOF does not directly provide an uncertainty analysis for the model; instead, this measure is available through postprocessing steps. Beckers et al. (2006) developed uncertainty metrics for DINEOF based on an analogy with optimal interpolation at a cost comparable to the interpolation itself. In contrast, DINCAE provides a reconstruction uncertainty. However, this uncertainty does not account for the inherent uncertainty in the satellite product. While some

adjustments are provided to adapt to the input uncertainty (Barth et al., 2022), these techniques remain quite limited. Further advancements are necessary in this area, enabling the current method's direct applicability for more extensive uncertainty analysis. Consequently, there is a need for continued development to fully integrate per-pixel uncertainty from the original satellite product into reconstruction models. Recent developments have focused on incorporating per-pixel uncertainty from the original satellite dataset into DINCAE by scaling the input using the inverse of the per-pixel error variance. While this

method shows potential, further evaluation is required to assess its effectiveness and implications.

## 4 Conclusion and outlook

The high missing rates of satellite-derived ocean colour products pose significant challenges to advancing our understanding of biogeochemical processes. Gap-filling these datasets is crucial for creating complete representations of phytoplankton dynamics. In this study, two gap-filling methods were applied for the first time to TChla and five major PFTs (diatoms,

dinoflagellates, haptophytes, green algae, and prokaryotes) over three years (2016–2019) along a corridor of a transect in the Atlantic Ocean, surveyed during the PS113 RV *Polarstern* expedition in 2018 with extensive in situ validation data (Bracher et al., 2020a). The first method, DINEOF, uses dominant empirical orthogonal functions, while the second, DINCAE, employs a convolutional autoencoder for reconstruction. A random search approach was used for hyperparameter optimisation.

DINEOF achieves roughly double the RMSLE in the degree of smoothing compared to DINCAE. The performance evaluation on the test dataset further highlights DINCAE's advantage, with DINEOF yielding RMSLE values that are 66 % higher for TChla, 11 % higher for diatoms, 20 % higher for dinoflagellates, 16 % higher for green algae, and 12 % higher for prokaryotes than DINCAE. Additionally, the MAPE results are consistent with the RMSLE findings. These errors vary significantly by location and group, with higher errors near continental shelves or areas with high missing data rates (31.7 %

for DINEOF and 26 % for DINCAE on total average error) and notably lower errors in the open ocean and oligotrophic regions (12.9 % for DINEOF and 9.7 % for DINCAE on total average error). Overall, DINCAE shows better gap-filling performance with test dataset validations, particularly in regions with complex water dynamics. External validation using in situ measurements reveals reduced accuracy in gap-filled data compared to the original dataset, with both models showing similar trends with increased dispersion. External validation indicates that DINEOF outperforms DINCAE, achieving better

regression performance, with approximately 12.5% improved slope, 13.6% improved intercept, and 68% higher R² for PFTs. In some cases, DINEOF even surpasses the original satellite dataset validation results. The MedPD, RMSD, and NRMSD



values are comparable between the two gap-filling methods but show variable performance relative to the original satellite data validation. Overall, external validation indicates that DINEOF is more suitable for large-scale gap-filling of ocean colour products, particularly for TChla and Chla of PFTs.

Test dataset performance evaluation and external validation results indicate that both methods demonstrate sufficient capability to reconstruct gaps within the ocean colour datasets. DINEOF is recommended for larger areas, open oceans and less complex waters, where it outperforms DINCAE in independent validation, offering the benefits of simpler tuning, lower computational costs, and more interpretable phenomena representation through EOF patterns. Although DINCAE offers higher accuracy across all areas in test dataset validation, its more complex architecture, demanding tuning procedure, and

requirement for GPU resources make it better suited for complex waters and coastal regions where precise reconstruction of original transient-scale patterns is critical. Future research should focus on a detailed uncertainty estimate for the reconstructed products, a critical improvement for their use in data fusion or assimilation. There are many other applications for the gap-free TChla and PFT data envisaged for marine ecosystem studies at regional and global scales. For example, detailed process studies rely mostly on a set of different but coincident in-situ measurements of physical and biogeochemical

parameters obtained during specific expeditions transecting regions of interest. The complete reconstruction of the regional phytoplankton community phenology using the gap-free satellite TChla and PFT Chla before, during and after research expeditions enhances tremendously the ability to link the different in-situ point measurements to each other. The gap-filled data set can enhance near real time research expedition planning to find hotspots of certain phytoplankton blooms in case of persistent cloud cover, limiting the satellite observations. Further, these gap-free data can ease the evaluation of global

biogeochemical models representing PFTs (e.g., Bopp et al., 2013; Dutkiewicz et al., 2015; Gürses et al., 2023).

## 5 Code and data availability

The preprocessing, processing, and postprocessing codes for generating gap-filled satellite-derived TChla and PFTs Chla datasets    are    available    on    Zenodo    (https://doi.org/10.5281/zenodo.14905369)    and    GitHub (https://github.com/EhsanMehdipour/PFT_gapfilling). The gap-filled datasets generated using both gap-filling methods for

the duration of independent in-situ measurements from the RV Polarstern PS113 expedition (May 10 to June 9, 2018) are available on Zenodo (https://doi.org/10.5281/zenodo.14905558). However, the complete gap-filled datasets spanning three years exceed 100GB in size and are stored on our server. These datasets can be provided upon request. The Source code for the DINEOF gap-filling model is available at https://github.com/aida-alvera/DINEOF (last accessed 15 March 2023). The source code for the DINCAE gap-filling model is available at https://github.com/gher-uliege/DINCAE.jl (last accessed 24

May 2024) or https://doi.org/10.5281/zenodo.5575066. The DPA-derived TChla and PFTs Chla concentrations, obtained from the pigment database, were published at https://doi.org/10.1594/PANGAEA.911061 Bracher et al. (2020b) and https://doi.org/10.1594/PANGAEA.954738 (Xi et al., 2023a). The original satellite-derived TChla and PFTs Chla concentrations are available from the Copernicus Marine Service website, as detailed in Sect. 2.1.2. The TChla and PFTs



Chla dataset can be accessed at https://doi.org/10.48670/moi-00280, and the SST dataset is available at
https://doi.org/10.48670/moi-00165.

## 6 Author contributions

The authors' contributions, outlined according to the Contributor Roles Taxonomy (CRediT) system, are as follows: **Ehsan Mehdipour**: Conceptualization, Data curation, Formal analysis, Investigation, Methodology, Software, Visualization, Writing – original draft preparation, Writing – review & editing; **Hongyan Xi**: Data curation, Formal analysis, Investigation,
Writing – review & editing; **Alexander Barth**: Methodology, Software, Writing – review & editing; **Aida Alvera-Azcárate**: Methodology, Software, Writing – review & editing; **Adalbert Wilhelm**: Conceptualization, Funding acquisition, Project administration, Supervision, Writing – review & editing; **Astrid Bracher**: Conceptualization, Data curation, Funding acquisition, Project administration, Supervision, Writing – review & editing.

## 7 Competing interests

The authors declare that they have no conflict of interest.

## 8 Acknowledgements

This study has been conducted using E.U. Copernicus Marine Service Information; https://doi.org/10.48670/moi-00280 and https://doi.org/10.48670/moi-00168. We thank ESA, EUMETSAT, and NASA for the ocean colour satellite data and the Copernicus Marine Service for the level 3 merged TChla and PFT products. We further acknowledge the captain, the crew,
and other scientists on board PS113 for their valuable support on board. We sincerely thank the reviewers for their thorough evaluation of the manuscript and their valuable and thoughtful suggestions. All text in this study was written by the co-authors, with AI assistance, such as ChatGPT, used solely to enhance the manuscript's readability and language.

## 9 Financial support

EM's contribution was part of the 4D-Phyto project, funded by AWI-INSPIRES and the Helmholtz School for Marine Data
Science (MarDATA) (Grant No. HIDSS-0005). HX's contribution was supported via the Copernicus Marine Service Evolution project GLOPHYTS (21036L05B-COP-INNOSCI-9000) and ML-PhyTAO (23138L03D-COP-INNO-SCI-9000) implemented by Mercator Ocean International. AAA's contribution was supported via the Copernicus Marine Service Evolution project MultiRes. Funding for RV *Polarstern* expedition PS113 data collection was supplied by the Helmholtz Infrastructure Initiative FRAM and ship time was provided under grant AWI_PS113_00.



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
