# Peer review of "Assessment of gap-filling techniques applied to satellite phytoplankton composition products for the Atlantic Ocean"

_EGUsphere, 2025_

## Author Comment (AC4)

**Assessment of gap-filling techniques applied to satellite phytoplankton composition products for the Atlantic Ocean**

Ehsan Mehdipour1,2, Hongyan Xi1, Alexander Barth3, Aida Alvera-Azcárate3, Adalbert Wilhelm2, Astrid Bracher1,4

Correspondence to: Ehsan Mehdipour (ehsan.mehdipour@awi.de)

**Author Comments in response to Referee #2**

The study entitled "Assessment of gap-filling techniques applied to satellite phytoplankton composition products for the Atlantic Ocean" by Mehdipour et al. addresses the critical challenge of missing data in satellite-derived ocean color products, particularly phytoplankton functional types (PFTs), which are crucial for understanding marine biogeochemical cycles and climate change impacts. The authors evaluate two established gap-filling methods, DINEOF and DINCAE, for reconstructing these datasets in the Atlantic Ocean. The manuscript is well written, interesting to read, the methodology sounds valid, and results and conclusions seem to be reasonable. I only have minor comments and suggestions for further improving the manuscript.

- We sincerely thank the reviewer for the thoughtful and constructive feedback, which provided valuable insights and helped us further improve the manuscript. We have carefully considered all comments and suggestions and have addressed each point in detail below. The corresponding revisions will be incorporated into the manuscript as indicated in our responses.
- 1. Page 8: the last two paragraphs are exactly the same (repeated).
  - The duplicated paragraphs on page 8 have been removed.
- 2. DINEOF and DINCAE are different in terms of their input data. For DINCAE, only three days of data (previous and next day) are used to reconstruct the missing pixels. However, the DINEOF processing needs data of much longer time series as input. From Section 2.3.1, it is not clear how many days of data are used for input. I suggest that this important information included in the description. Also, please discuss the effect of the length of the time series on the performance of the two models.
  - The number of days of data used for reconstruction is indeed a hyperparameter that we tuned during model development. In DINEOF, although the full dataset is available for extracting EOF values, the effective temporal covariance is influenced by the Laplacian filtering parameters ( $\alpha$  and P). The extent of this filter, calculated as L =  $2\pi\sqrt{\alpha p}$ , was approximately 5.74 days in our final model, indicating that temporally close datasets (within about six

<sup>1Alfred Wegener Institute (AWI), Helmholtz Centre for Polar and Marine Research, Bremerhaven, Germany

<sup>2School of Business, Social & Decision Sciences, Constructor University, Bremen, Germany

<sup>3GeoHydrodynamics and Environment Research (GHER), University of Liège, Liège, Belgium

<sup>4Institute of Environmental Physics, University of Bremen, Bremen, Germany

days) have the greatest influence during reconstruction (see Section 2.3.3). For DINCAE, several window sizes (3, 5, and 7 days) were tested during hyperparameter tuning of the ntime\_win parameter. The model using a 3-day window achieved the best cross-validation performance and was therefore selected for the final reconstruction. These details are already included in the text.

- A new statement has been included in Section 3.5 (Novelty and limitations) discussing the impact of time series length on the performance of the two models (see lines 692-701 of the revised manuscript).
  - "Furthermore, the length of the time series can significantly influence the performance of both models. A short time series may result in EOF patterns that do not fully capture the underlying variability in the dataset when using the DINEOF method, and it may also lead to suboptimal tuning of parameters and hyperparameters in DINCAE due to limited training samples. A long time series primarily affects the DINEOF method, as EOF extraction over extended periods tends to emphasize more persistent and large-scale spatial patterns while reducing the representation of transient variability. This effect is less pronounced in DINCAE, since the data are temporally segmented into minibatches before being processed by the network, which decreases, but does not eliminate, its dependence on the total record length. When the time series is extended, the corresponding increase in training epochs allows the model to learn from a broader range of examples and improves its generalization ability, but it may also lead to a reduced sensitivity to short-lived or rare features. Although not examined in the present study, the length of the time series could be considered an experimental hyperparameter to be optimized during model development."
- 3. Equation (1) is confusing: t is not defined, SST is not included, T is not specified, X-CHL is repeated for T times?
  - We have reformulated Equation (1) to improve clarity. All parameters, including t and T, are now explicitly defined in the text, and SST has been incorporated into the formulation. Additionally, the notation X1CHL now clearly represents the total chlorophyll-a concentration dataset for day 1. The revised equation and its description can be found in Section 2.3.1 of the manuscript.
- 4. Section 3.2 and Fig. 5-6: Please discuss the effect of the data miss rate on the model performance in terms of different parameters (TChl vs. PFTs), and spatial regions (high latitude, equatorial Vs. mid-latitude).
  - It is challenging to directly compare the effect of data availability across different spatial regions (e.g., high-latitude, equatorial, and mid-latitude areas), since the physical and biological dynamics in these regions are inherently distinct. This variability makes it difficult to isolate the impact of data availability from the influence of regional dynamics and create a one-to-one comparison. To investigate the relationship between data availability and model performance across different groups, we generated 3D histograms showing the relationship between data availability and the mean absolute error (MAE) for each pixel in the test dataset (Figures S3 and S4, shown at the end of this document). These plots illustrate how model performance (in terms of MAE) varies with changes in data availability. Both models show that higher data availability generally corresponds to lower and more tightly clustered MAE values, indicating improved reconstruction accuracy. In particular, the MAE for TChla decreases as data availability increases. When comparing the two models, DINCAE consistently produces reconstructions with lower MAE than DINEOF, suggesting that DINCAE more effectively captures the underlying spatio-temporal variability, especially in regions with limited data availability.
  - These figures were not added to the manuscript, but a short statement referring to this work was added:

- "In addition, it is challenging to isolate the effect of data availability on model performance across different spatial regions (e.g., high-latitude, equatorial, and mid-latitude areas), since the physical and biological dynamics in these regions are inherently distinct. The investigation of the relationship between data availability and gap-filling model performance across different groups based on mean absolute error (MAE) (results not shown) showed, for both models, as expected, that a higher data availability generally corresponds to lower and more tightly clustered MAE values, indicating improved reconstruction accuracy. When comparing the two models, DINCAE consistently produces reconstructions with lower MAE than DINEOF, suggesting that DINCAE more effectively captures the underlying spatiotemporal variability, especially in regions with limited data availability."
- 5. Page 23, line 557: change VIIRS-SNAP to VIIRS-SNPP
  - The term "VIIRS-SNAP" has been corrected to "VIIRS-SNPP" in the revised manuscript.
- 6. It is noted that the PFTs shows significant difference with in situ measurement, but they are still valuable datasets. Further improvement of the quality of the PFT data are expected in the future.
  - We agree that although the current PFT and gap-filled PFT products show discrepancies compared to in situ measurements, they remain valuable for large-scale and long-term analyses. We have emphasized this point in the conclusion section, highlighting the limitations of the current PFT datasets and the need for future improvements in retrieval accuracy and data quality. The revised text can be found in Section 4, Conclusion and Outlook, and the following statement was added:
    - "Although the current methods are best suited for regional monitoring, scaling them to global applications would require additional optimization. Nevertheless, constructing a globally gap-filled Chla dataset, even at a reduced spatiotemporal resolution, could provide invaluable input for long-term climate assessments, global biogeochemical modeling, and validation of Earth system models."

Figure S3. Histogram distribution of the test dataset's MAE vs Data Availability for DINCAE gap-filling

Figure S4. Histogram distribution of the test dataset's MAE vs Data Availability for DINEOF gap-filling